# Mitigating Heterogeneous Token Overfitting in LLM Knowledge Editing

**Tianci Liu**[1]  **Ruirui Li**[2]  **Zihan Dong**[3]  **Hui Liu**[2]  **Xianfeng Tang**[2]  **Qingyu Yin**[2]  **Linjun Zhang**[3]  **Haoyu Wang**[4]
**Jing Gao**[1]

## Abstract

Large language models (LLMs) have achieved remarkable performance on various natural language tasks. However, they are trained on static corpora and their knowledge can become outdated quickly in the fast-changing world. This motivates the development of knowledge editing (KE) to update specific knowledge in LLMs without changing unrelated others or compromising their pre-trained capabilities. Previous efforts sought to update a small amount of parameters of a LLM and proved effective for making selective updates. Nonetheless, the edited LLM often exhibits degraded ability to reason about the new knowledge. In this work, we identify a key issue: *heterogeneous token overfitting* (HTO), where the LLM overfits different tokens in the provided knowledge at varying rates. To tackle this, we propose OVERTONE, a token-level smoothing method that mitigates HTO by adaptively refining the target distribution. Theoretically, OVERTONE offers better parameter updates with negligible computation overhead. It also induces an implicit DPO but does not require preference data pairs. Extensive experiments across four editing methods, two LLMs, and diverse scenarios demonstrate the effectiveness and versatility of our method.

## 1. Introduction

Language models (LMs) parameterized by deep neural networks (Vaswani et al., 2017; Lewis et al., 2019; Radford et al., 2019; Brown et al., 2020) demonstrate strong generalizability across various natural language generation and classification tasks (See et al., 2019; Raffel et al., 2020; Ji et al., 2023). These successes underscore their

versatility, establishing them as new foundations for natural language processing applications (Bommasani et al., 2021; Zhou et al., 2023). Furthermore, with model sizes continually increasing, large language models (LLMs) exhibit emerging abilities to follow natural language instructions (Dong et al., 2022b; Ouyang et al., 2022), which empowers their zero-shot adaptations to unseen tasks (Kojima et al., 2022), paving the way towards artificial general intelligence (Bubeck et al., 2023).

Despite this remarkable potential, the real-world LLM deployment remains largely unresolved: LLMs are capable of comprehending a wide range of human instructions and queries, but they can only provide feedback based on their *static* knowledge from the data they were trained on. In a fast-changing world, most knowledge quickly becomes outdated. For example, the updated knowledge about *the president of United States* would refer to *Donald Trump* rather than *Joe Biden*. Failing to maintain update-to-date knowledge could amplify critical issues such as making factual fallacy (De Cao et al., 2021) or producing harmful generations (Hartvigsen et al., 2022). However, the significant computational cost of retraining makes it impractical to frequently incorporate new knowledge.

As a remedy, *knowledge editing* (KE), whose goal is to update an LLM with some *specific* knowledge without hurting irrelevant others and general ability, is proposed (Wang et al., 2023b; Zhang et al., 2024c). Full fine-tuning of LLMs proved ineffective as it severely disrupted irrelevant knowledge (Wang et al., 2023b), leading to an *editing-locality* trade-off. Here *locality* refers to the ability to maintain knowledge unrelated to the update, such as *the prime minister of Canada* for the previous case. To achieve a good locality, model updates need to be *selective* and should rely on *a small fraction* of parameters (Wang et al., 2023b). Following this principle, parameter-efficient fine-tuning (PEFT) methods such as LoRA (Hu et al., 2021) have achieved good performance (Wu et al., 2023). On the other hand, Huang et al. (2023); Dong et al. (2022a) restricted the updates to some pre-specified feed-forward network (FFN) layer that serves as knowledge storage (Dai et al., 2021). Meng et al. (2022a;b) refined the process by introducing a *locating* stage to identify which layer the target knowledge is stored. These fine-grained manners have demonstrated impressive success

[1]Purdue University [2]Amazon [3]Rutgers University [4]University at Albany. Correspondence to: Haoyu Wang <hwang28@albany.edu>, Jing Gao <jinggao@purdue.edu>.

*Proceedings of the 42nd International Conference on Machine Learning*, Vancouver, Canada. PMLR 267, 2025. Copyright 2025 by the author(s).

in maintaining high locality (Zhang et al., 2024c).

Nevertheless, existing methods still suffered from losing LLM generalizability, especially when dealing with tasks that involve the edited knowledge, due to the so-called *overfitting* of KE (Zhang et al., 2024a). Specifically, KE often involves one piece of new knowledge to edit at a time, which entails updating (selected) parameters with single training instance. Consequently, edited LLMs tend to pay excessive attention to the edited subject, but fail to reason about the new knowledge (Zhong et al., 2023; Zhang et al., 2024a). Previous works highlighted this challenge, and quantified this ability with a new metric known as *portability* (Zhong et al., 2023; Wang et al., 2024f). However, the underlying causes of overfitting and their relationship to the KE process remain under-explored, leaving *if KE overfitting can be solved in a principled manner* an open question.

In this work, we take the first step toward a deeper understanding of this overfitting, and pave the way for a principled solution to mitigate it. We first provide strong evidence that *KE overfitting leads to catastrophic degradation of an LLM's reasoning ability*. In particular, we showed that as the LLM is edited with new knowledge, the probability of correct reasoning consistently decreases. To quantify this, we investigated the *portability loss* at each fine-tuning step (lower indicates better reasoning ability). We observed that while portability loss initially decreased, it grew up quickly thereafter. In addition, the final loss was significantly higher than the initial value. This finding confirms that overfitting is a direct cause of suboptimal portability.

To understand this overfitting, we checked how new knowledge is fitted during the KE process. Based on our findings, *KE may only require learning a few pivotal tokens (words)*, as many tokens already exhibit small *initial* loss values. Intuitively, an LLM's pre-trained knowledge may enable it to *infer* remaining parts base on pivotal tokens. However, existing methods overlook this token-level difference in KE. Even when selectively updating parameters, these methods aim to maximize the likelihoods of the entire sentence describing the new knowledge, which boils down to maximizing the probability of *all* tokens indiscriminately (Bengio et al., 2000; Radford et al., 2019; Brown et al., 2020). As a result, this coarse-grained training paradigm leads to varying degrees of overfitting across tokens. We term this phenomenon *heterogeneous token overfitting* (HTO) in KE. Sec 2 details our new insight on KE overfitting and its influence on portability. *This is our first main contribution.*

In light of how HTO roots at a token level, we propose OVERTONE, a new KE training paradigm to tackle it. OVERTONE assigns each token an adaptive training target according to its (over)fitting state. An efficient solution is proposed to construct these training objectives in a dynamic way that allows to maintain much pre-trained knowledge if

possible. The theoretical advantage of our method lies in three folds. First, our solution induces negligible computation cost compared to standard training (much cheaper than a LLM forward). Second, our solution provides a better parameter update through the lens of importance function (Koh & Liang, 2017). Finally, OVERTONE has a close connection to direct preference optimization (DPO), a widely-used framework for LLM post-training (Rafailov et al., 2024; Zhang et al., 2024d), but does not require additional preference data pairs. Sec 3 covers these aspects in details. *The proposed OVERTONE and our theoretical analysis is another main technical contribution of this work.* Remarkably, OVERTONE can be of interest to other tasks such as machine unlearning, where selective updates of LLMs are desired. Moreover, when the training text is long, as the number of tokens to learn grows, we expect HTO to exacerbate, and OVERTONE to be helpful.

Our paper is organized as follows. Sec 2 and Sec 3 details the new overfitting phenomenon in KE and our proposed OVERTONE for mitigation respectively. Extensive experimental results in Sec 4 demonstrate the superiority of our solution. In the remaining part of this paper, we review related works in Sec 5, and conclude the paper in Sec 6.

## 2. Overfitting Issue in Knowledge Editing

This section presents a new token-dependent overfitting phenomenon in knowledge editing (KE) that has been overlooked in the literature. Background of KE is also provided.

### 2.1. Preliminaries

Given a text $\boldsymbol{x} = (x_1, \ldots, x_n)$, where each $x_i \in \mathcal{V}$ is a token from vocabulary $\mathcal{V}$, a large language model (LLM) parameterized by $\theta$ computes probability $\pi_\theta(\boldsymbol{x})$ based on chain rule (Bengio et al., 2000):

$$\pi_\theta(\boldsymbol{x}) = \prod_{i=1}^{n} \pi_\theta(x_i \mid x_1, \ldots, x_{i-1}) \triangleq \prod_{i=1}^{n} \pi_\theta(x_i \mid \boldsymbol{x}_{<i}),$$

where $\pi_\theta(x_i \mid \boldsymbol{x}_{<i})$ is the predicted distribution of token $x_i$ given previous $\boldsymbol{x}_{<i}$. The LLM is usually trained with maximum likelihood estimation (Hochreiter, 1997; Sutskever, 2014; Cho et al., 2014). To generate a sentence $\boldsymbol{x}$, the LLM computes $\pi_\theta(x_i \mid \boldsymbol{x}_{<i})$ and draws $x_i$ from it; then $x_i$ is combined with $\boldsymbol{x}_{<i}$ as new inputs for future steps. This process completes if a special token that marks the end of the sentence is returned, or if the maximum length is reached.

**Knowledge Editing (KE)** aims to update specific knowledge in a pre-trained LLM while preserving unrelated others. A knowledge can be represented by natural language $(\boldsymbol{x}, \boldsymbol{y})$, $\boldsymbol{x}$ describes the *subject* and *relation*, and $\boldsymbol{y}$ entails corresponding *object*. For instance, suppose $\boldsymbol{x}$ is *The president of United States is*, $\boldsymbol{y}$ can be *Donald Trump*. KE asks the

LLM to respond given $x$ with new $y$, while satisfying the following criteria meanwhile (Zhang et al., 2024c): (1) **Generality:** the edited model should generalize to all equivalent inquires about the *US president*. (2) **Portability:** questions reasoned from the new knowledge such as *the first lady of United States* should be answered correctly. (3) **Locality:** unrelated knowledge such as *the prime minister of Canada* should be unchanged. These requirements of precisely updating specific knowledge proves non-trivial (Wang et al., 2023b; Zhang et al., 2024c).

## 2.2. Overfitting in Knowledge Editing

In response to precise KE requirements, existing attempts restrict the updates to only a minimal amount of parameters. This design establishes remarkable progress in maintaining good locality (Zhang et al., 2024c; Wang et al., 2024d). However, it proves insufficient to maintain good generalizability (generalilty and portability) due to the so-called *overfitting* issue (Zhong et al., 2023; Zhang et al., 2024a).

Namely, many KE tasks involve one piece of new knowledge at a time, requiring to fine-tune an LLM on single training instance. In such challenging scenarios, the LLM often encounters severe overfitting even only a few parameters are updated. This greatly restricts its ability to generalize the edited knowledge. As shown in Zhong et al. (2023); Zhang et al. (2024a), edited LLMs usually pay excessive attention to the edited subject, but fail to address multi-hop reasoning questions involving the new knowledge. As a result, this limitation results in suboptimal portability.

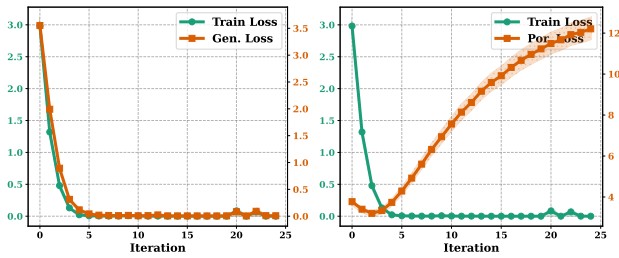

*Figure 1.* Loss (average) change of ground truth answers to *generality* (rephrased, left) and *portability* (reasoning, right) questions.

As a direct evidence, Fig 1 shows the change of generality and portability loss[1] at different iterations from fine-tuning LLaMA2 7B (Touvron et al., 2023) with LoRA, a representative KE baseline method (Zhang et al., 2024c). As the training goes on, the generality loss decreases. However, the portability loss decreases at the beginning of training, but starts to increase later. This confirms the existence of overfitting. More importantly, the ultimate portability loss is significantly larger than before editing, indicating that *the reasoning ability is in fact undermined by the KE process*,

---

[1]The perplexity loss of the ground truth answer to a question.

## 2.3. Heterogeneous Token Overfitting

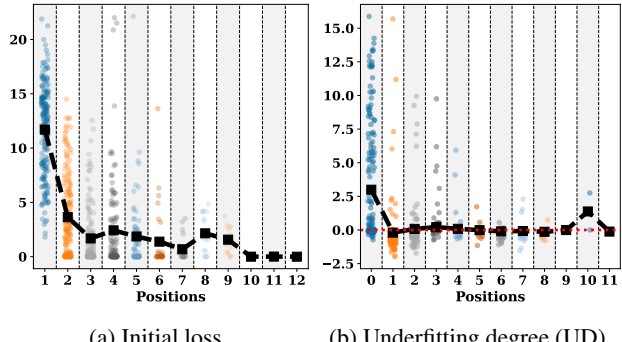

(a) Initial loss.      (b) Underfitting degree (UD).

*Figure 2.* Token-level initial loss and UD (negative indicates over-fitted). Dashed lines mark the mean values.

Towards a deeper understanding of this overfitting phenomenon, we check the loss of each token, and find that *different tokens tend to have distinct initial loss values*. As depicted in Fig 2a, before editing LLaMA2, only certain tokens (e.g., the beginning) have significant loss values. On the other hand, some tokens take small loss value and are *initially-fitted* by nature. As an intuitive explanation, consider the previous *US president* example. No matter a user wants to edit the answer to *Donald Trump* or *Joe Biden*, after seeing the first word *Donald* or *Joe* as a *hint*, the LLM is expected to be capable of infer the remaining part based on its pretrained knowledge.

Nonetheless, existing KE methods overlook this token-level difference. Consequently, they tend to overfit tokens that have varied losses at different speeds. For verification, we compute the *pre-edited* log-likelihood of tokens generated by the model with greedy decoding, and that of the editing instance during the KE process. Note that our choice of greedy decoding is on purpose, as it reflects the unedited model's most confident knowledge proper that was valid in the past. By comparing the loss of the two, we can measure if a token is overfitted. Specifically, we define *underfitting degree* (UD) as the difference between the pre-edited and running log-likelihoods. Here negative UD indicates an overfitting. Fig 2b shows UD of different tokens when half of them are overfitted. Strong pattern of UD varies across different tokens confirms our concern. We dub this issue as *heterogeneous token overfitting* (HTO) of KE.

HTO's direct cause lies in the training paradigm. Formally, given editing instance $(\boldsymbol{x}, \boldsymbol{y} = [y_1, \ldots, y_m])$ where $\boldsymbol{y}$ contains $m$ tokens, many KE methods resort to a conventional LLM training objective[2]. In particular, they seek to maximize likelihood of $\pi_\theta(\boldsymbol{y} \mid \boldsymbol{x})$ by minimizing an *averaged*

---

[2]We restrict our study to the widely-used *teacher-forcing* mechanism (Lamb et al., 2016).

cross-entropy (CE) loss with gradient descent on

$$\ell_{\text{CE}}(\theta) \triangleq \sum_{i=1}^{m} \text{CE}[\delta_{y_i}(y) \| \pi_\theta(y \mid \boldsymbol{x} \oplus \boldsymbol{y}_{<i})] \quad (1)$$

$$= -\sum_{i=1}^{m} \log \pi_\theta(y_i \mid \boldsymbol{c}_i)$$

$$\nabla_\theta \ell_{\text{CE}}(\theta) = -\sum_{i=1}^{m} \nabla_\theta \log \pi_\theta(y_i \mid \boldsymbol{c}_i).$$

Here $\boldsymbol{c}_i = \boldsymbol{x} \oplus \boldsymbol{y}_{<i}$ denotes the context for token $y_i$, $\delta_{y_i}(y)$ is the Kronecker delta function[3], and $\text{CE}[\cdot\|\cdot]$ computes CE between two distributions.

During training, gradient $\nabla_\theta \ell_{\text{CE}}(\theta)$ maximizes the probability of $y_i$ whiling minimizing the probabilities of all other candidates. When the model is repeatedly updated using gradient(s) from the *single* datapoint, as in KE, the probabilities of *initially-fitted* tokens become disproportionately large, while tokens with high initial loss values are gradually fitted. That is to say, HTO lies in *indiscriminately* optimizing CE loss of *all* tokens, without considering their difference. Existing attempts for mitigating overfitting such as early stopping (Yao et al., 2007) and label smoothing (Szegedy et al., 2016; Müller et al., 2019) also ignore this token-level difference, making them conceptually less suitable for HTO.

## 3. Propose Method

Given the importance of token-level difference in HTO, we propose OVERTONE to offer a granular control that applies to various KE methods, theoretical analysis is also provided.

### 3.1. Counteract HTO with OVERTONE

We present OVERTONE, a token-level strategy for HTO mitigation. Our method *smooths* $\boldsymbol{y}$'s distribution for fitting in an adaptive way. Specifically, we replace each delta distribution $\delta_{y_i}(y)$ with a unique smoothed *target* distribution $\pi_{\text{tar}}(y \mid \boldsymbol{c}_i)$, and refine the cross entropy by a clipped *forward* KL divergence. Our complete loss is given by

$$\ell_{\mathcal{O}\text{VERTONE}}(\theta) \triangleq \sum_{i=1}^{m} \max(\text{D}_{\text{KL}}[\pi_{\text{tar}}(y \mid \boldsymbol{c}_i)\|\pi_\theta(y \mid \boldsymbol{c}_i)], \epsilon),$$
$$(2)$$

where clipped $\max(\cdot, \epsilon)$ imposes a *token-level* early stopping when predicted $\pi_\theta$ is close enough to $\pi_{\text{tar}}$.

**Principles of $\pi_{\text{tar}}$ design.** We note that two principles should be met in order to make $\pi_{\text{tar}}$ a good distribution to target on. First, $\pi_{\text{tar}}$ should convey that ground truth token $y_i$ is most probable, otherwise, the objective may lead

___
[3] $\delta_{y_i}(y) = 1$ if $y = y_i$ else 0.

to incorrect knowledge. Second, compared to uniform prior that smooths all tokens equally, the model's own pre-trained knowledge is a better prior to help mitigate forgetting problem (Zhang & Sabuncu, 2020; Lee et al., 2022).

In light of the two principles, we use $\delta_{y_i}$ and the LLM's *current* knowledge from its predicted distribution $\pi_\theta$ to construct target $\pi_{\text{tar}}$. However, as will be verified later, directly use $\pi_\theta$ can be suboptimal due to the non-negligible noise it carries (Hewitt et al., 2022; Tang et al., 2024). Specifically, Tang et al. (2024) argued that $\pi_\theta$ mixes a distinct subset of *informative* tokens, and a subset of *noisy* tokens associating with small *logits* that fall outside $n\sigma$-distant away from the maximal value. By filtering out noisy tokens in $\pi_\theta$, the LLM performance can be boosted at inference time. We bring this insight to the training (editing) phase and mix the *filtered* distribution[4] $\pi_{\text{flt}}^{(i)}$ with $\delta_{y_i}$ by

$$\pi_{\text{tar}}^{(i)} \triangleq \begin{cases} \pi_{\text{tar}}^{\text{can}} \triangleq \lambda\delta_{y_i} + (1-\lambda)\pi_{\text{flt}}^{(i)} & \text{if } y_i = \text{argmax}_y \pi_{\text{tar}}^{\text{can}}, \\ \delta_{y_i} & \text{otherwise}, \end{cases}$$
$$(3)$$

where $\lambda$ is a hyper-parameter. Namely, we adopt the candidate mixture $\pi_{\text{tar}}^{\text{can}}$ if it correctly assigns the maximal probability to $y_i$, otherwise, we *skip* the mixing and use $\delta_{y_i}$. This *skip* mechanism helps reduce potential knowledge conflicts by discarding $\pi_{\text{flt}}^{(i)}$ (from $\pi_\theta$) when it heavily relies on outdated knowledge, which often happens in the first few training steps, empirical benefit is shown in Sec 4.4. Algo 1 outlines the process of our solution.

### 3.2. Theoretical Advantages of OVERTONE

This section provides theoretical analysis on key factors that merit OVERTONE for KE. All proofs and more in-depth technical background are deferred to App A.

**Merit 1. OVERTONE is universal and efficient.**

While seemingly distinct, OVERTONE is in fact a generalization of CE loss. Moreover, our choice of $\pi_{\text{tar}}$ makes it computationally efficient, with computation overhead negligible compared to LLM forward operation.

**Proposition 3.1.** *OVERTONE loss generalizes CE loss and reduces to the latter when $\epsilon = 0, \lambda = 1$.*

**Proposition 3.2.** *Using Alg 1, the additional computation complexity induced by OVERTONE is $\mathcal{O}(|\mathcal{V}|)$ when fitting a token, where $|\mathcal{V}|$ is the vocabulary size.*

**Merit 2. OVERTONE provides better updates.**

OVERTONE leads to more effective parameter updates, as demonstrated through the lens of the influence function (Koh

___
[4] For brevity $\pi_{\text{flt}}^{(i)} = \pi_{\text{flt}}(y \mid \boldsymbol{c}_i)$, $\pi_{\text{tar}}^{(i)}$ is defined similarly. Plain $\pi_{\text{flt}}$ and $\pi_{\text{tar}}$ will be used when discussing the general idea.

**Algorithm 1** OVERTONE Training Paradigm

---

1: **Input:** Editing data $(\boldsymbol{x}, \boldsymbol{y} = [y_1, \ldots, y_m])$, LM parameters $\theta_0$, mixing hyper-parameter $\lambda$, early-stopping threshold $\epsilon$, filtering threshold $n$, total training steps $T$.

2: **Initialize:** $\theta = \theta_0$.

3: **for** $t = 1, \ldots, T$ **do**

4:     *# Inner loop is parallelized in practice, unroll for better readability.*

5:     **for** $i = 1, \ldots, m$ **do**

6:         Set context $\boldsymbol{c}_i = \boldsymbol{x} \oplus \boldsymbol{y}_{<i}$.

7:         Compute logits from the LM as $\boldsymbol{s}^{(i)} = f_\theta(\boldsymbol{c}_i) \in \mathbb{R}^{|\mathcal{V}|}$. Take softmax and get $\pi_\theta^{(i)}$.

8:         Top $n\sigma$-filter (Tang et al., 2024): Compute $s_{\max}^{(i)} = \max_k \boldsymbol{s}^{(i)}$, $\sigma = \text{std}(\boldsymbol{s}^{(i)})$. Define filtered logit $\tilde{s}_k^{(i)} = -\infty$ if $s_k^{(i)} \leq s_{\max}^{(i)} - n\sigma$ else $\tilde{s}_k^{(i)} = s_k^{(i)}$.

9:         Take softmax on filtered $\tilde{\boldsymbol{s}}$ and get filtered $\pi_{\text{flt}}^{(i)}$.

10:        Compute target $\pi_{\text{tar}}^{(i)}$ based on Eq (3).

11:        Compute loss

$$\ell_{OVERTONE}^{(i)} = \max(D_{\text{KL}}[\pi_{\text{tar}}^{(i)} \| \pi_\theta^{(i)}], \epsilon).$$

12:     **end for**

13:     Compute sample loss

$$\ell_{OVERTONE}(\theta) = \sum_{i=1}^{m} \ell_{OVERTONE}^{(i)}.$$

14:     Update with learning rate $\alpha$

$$\theta \leftarrow \theta - \alpha \nabla_\theta \ell_{OVERTONE}(\theta)$$

15: **end for**

**output** Edited parameter $\theta$.

---

& Liang, 2017), outlined in the following informal theorem. Due to page limitations, the formal version and corresponding assumptions are deferred to Appendix A.3.

**Theorem 3.3** (Informal). *Under regularity conditions, compared to optimizing the vanilla CE loss, OVERTONE provides a more favorable update direction for the parameters and has less influence on unrelated knowledge.*

**Merit 3. OVERTONE has close connection to DPO and other constrained optimizations.**

One might question whether OVERTONE is conceptually superior to constrained optimization approaches, such as fine-tuning only a small set of specific parameters (Dong et al., 2022a; Dai et al., 2021), limiting update magnitudes (Zhu et al., 2020), or employing low-rank updates (Hu et al., 2021). We emphasize that OVERTONE introduces a new objective that can be solved with any optimization methods, regardless of whether constraints are imposed. In other

words, OVERTONE can be seamlessly combined with existing constrained optimization-based solutions for KE.

Below theorem draws a connection between OVERTONE and direct preference optimization (DPO), which has shown superior performance of maintaining pretrained knowledge in LLM post-training (Wang et al., 2023a).

**Theorem 3.4.** *Let $\epsilon = 0$, optimizing OVERTONE can be seen as optimizing an unbiased estimate of a DPO objective plus some additional KL penalty.*

Compared with conducting explicit DPO, OVERTONE does not require collecting preference data, and is more efficient thereof. Furthermore, as highlighted in Rozner et al. (2024), another challenge of applying DPO to KE is that determining *win-loss* data pairs can be unstraightforward in KE. In contrast, OVERTONE walks around this challenge by refraining from treating any token as *unpreferred*, and instead acts on a distribution level.

# 4. Experiments

We evaluate the proposed OVERTONE paradigm on four performant KE methods applying to two representative large language models (LMs) over five benchmarking datasets. Ablation studies are also conducted to help understand its effectiveness. Results show that OVERTONE helps improve editing performance by a large margin on all methods. More conceptual discussions can be found in Appendix D.

### 4.1. Experiment Setup

**Base Models.** We conduct experiments on two representative LMs, **LLaMA 2-7b-Chat** (Touvron et al., 2023) and **LLaMA 3-8b-Instruct** (Dubey et al., 2024), which have been widely studied in the literature (Zhang et al., 2024c; Wang et al., 2024d). From now on, we refer to the two LMs as **LLaMA 2** and **LLaMA 3** for brevity.

**Tasks.** Following Wang et al. (2023b); Zhang et al. (2024c), we edit different kinds of knowledge: WikiData$_{\text{recent}}$, WikiData$_{\text{counterfact}}$ (Cohen et al., 2024), WikiBio (Hartvigsen et al., 2024), and ZsRE (Yao et al., 2023). Besides the four popular benchmarks, we also explore more complex MQuAKE (Zhong et al., 2023; Wang et al., 2024f). Due to page limitation, we refer readers to Zhang et al. (2024c) for more benchmark details. When editing an LLM, we consider two scenarios: (1) **Single Editing**: one piece of knowledge is edited at a time. (2) **Continual Editing**: multiple pieces of knowledge are edited in a sequential way. This is more challenging due to forgetting and knowledge conflicting (Hartvigsen et al., 2024; Wang et al., 2024d).

**Editing Methods.** We apply OVERTONE to four representative KE methods from different families that have achieved state-of-the-art performance (Zhang et al., 2024c; Wang

et al., 2024e). **FT-M** (Zhang et al., 2024c) fine-tunes a special layer identified by causal-tracing analysis wherein the knowledge is stored. **LoRA** (Hu et al., 2021) learns additive low-rank updates for model parameters on the new knowledge. **MELO** (Yu et al., 2024) and **WISE** (Wang et al., 2024d) incorporates additional parameter copies to learn new knowledge, along with some gating mechanism to determine whether original or new knowledge should be used at inference time. Despite incorporating certain explicit or implicit constraints on the learnable parameters, these methods are all trained to minimize the CE loss. For better benchmarking, we also report results from two widely-studied methods **ROME** (Meng et al., 2022a) and **MEMIT** (Meng et al., 2022b). ROME applies a causal-tracing analysis to identify the layer wherein the knowledge is stored and then solves an analytic rank-one update, and MEMIT extends ROME by identifying a series of layers to edit and finding the updates as least-squares solutions. *To reflect the challenging nature of KE under data scarcity regime, we focus on KE methods that do not require a larges-scale hard-to-access training data, or training additional models. No data augmentation were applied during the editing.*

**Evaluation Criteria.** We evaluate the performance from four aspects as discussed in Sec 2: **reliability (Rel.)**, **generality (Gen.)**, **portability (Por.)**, and **locality (Loc.)**. Due to page limits we refer readers to Zhang et al. (2024c); Wang et al. (2024d) for their formulations. We report the average of different metrics for more complete comparisons.

**Implementation Details.** All of our experiments are implemented in EasyEdit (Wang et al., 2024e). More details and hyper-parameters can be found in App B.

### 4.2. Single Editing Performance

We evaluate the effectiveness of OVERTONE in conducting Single Editing on ZsRE, WikiData_{recent}, WikiData_{counterfact}, and WikiBio with different KE methods. WISE was tested on ZsRE, the only benchmark that contains additional irrelevant data during the editing time that is required by WISE.

Single Editing results are reported in Tab 1. From the table, all KE methods gained significant improvement from the proposed OVERTONE paradigm. Specifically, The four methods hardly performed comparable to baselines ROME and MEMIT from normal training, but were capable of exceeding them when trained with OVERTONE. For instance, without OVERTONE, ROME achieved the highest and the second-highest average performance for editing LLaMA 2 and LLaMA 3 respectively on Wiki_{recent}. However, when equipped with OVERTONE, FT-M, LoRA, and MELO outperformed ROME on both tasks.

We next check where the improvement was made. From the table, the first gain was from improved portability. To

see this, note that when editing LLaMA 2 on ZsRE, LoRA reached a portability that was nearly three times of the base version. Similarly, MELO also reached an almost doubled portability. More evidence can be found from editing LLaMA 3 as well. In addition, all methods, especially those initially fall short in maintaining good locality, achieved excellent performance in this regard. As an evidence, LoRA's reached a nearly five times locality improvements when editing both LLaMA 2 and LLaMA 3 on Wiki_{counterfact}. We want to highlight that, all these improvements were made *without compromising editing reliability*. That is to say, all the four methods achieved better trade-offs between reliability and reasoning (and locality) from the proposed OVERTONE. More importantly, this success was established in a *model-agnostic* manner, in the sense that OVERTONE is not specialized for any particular KE method studied here. Instead, it offers a highly flexible and generic paradigm that can be combined with existing solutions in a plug-and-play manner.

**More Complex Editing task**. To further evaluate how OVERTONE performs on complex benchmark in the filed of KE, we test FT-M and LoRA with editing the two LLMs on MQuAKE-2002 (Wang et al., 2024f)[5], following Zhong et al. (2023). This task requires the edited LLM to answer single- and multi-hop reasoning questions about the edited knowledge. Experiment results are reported in Table 2. As before, OVERTONE was capable of achieving better portability without hurting the editing performance.

These empirical results echo well with our theoretical analysis, and confirm the superiority of OVERTONE.

### 4.3. Continual Editing Performance

We next study the more challenging scenarios, where massive edits are conducted in a continual (sequential) way. Experiments were again run on the four benchmarks.

Due to page limit, We defer the complete results to App C, and visualize the average of reliability, generality, portability, and locality in Fig 3. Specifically, we evaluate the performance after new $T$ pieces of knowledge length are edited sequentially. Different KE methods are represented in separate colors. Solid boxes indicate normal training performance, and transparent boxes show results from training with OVERTONE. The unfilled area within the boxes quantifies the improvements form OVERTONE.

As in Single Editing scenarios, OVERTONE again improved the performance of four KE methods, enabling them to surpass ROME and MEMIT by a large margin across diverse settings. Furthermore, on three out of the four benchmarks (ZsRE, Wiki_{recent}, and Wiki_{counterfact}), the improve-

---

[5]This is a cleaned version of MQuAKE by fixing knowledge conflicts (Wang et al., 2024f).

*Table 1.* Single Editing performance. Four KE methods gained improvement from OVERTONE training paradigm. WISE requires additional irrelevant data for training, which is only available in ZsRE benchmark.

| | ZsRE | | | | | Wiki$_{recent}$ | | | | Wiki$_{counterfact}$ | | | | WikiBio | | |
|---|---|---|---|---|---|---|---|---|---|---|---|---|---|---|---|---|
| | **LLaMA 2-7b-chat** | | | | | | | | | | | | | | | |
| | Rel. | Gen. | Por. | Loc. | Avg. | Rel. | Por. | Loc. | Avg. | Rel. | Por. | Loc. | Avg. | Rel. | Loc. | Avg. |
| **ROME** | 96.61 | 83.91 | 55.7 | 96.96 | 83.3 | 99.02 | 54.21 | 55.91 | 69.71 | 97.2 | 56.85 | 50.4 | 68.15 | 96.41 | 59.14 | 77.78 |
| **MEMIT** | 94.22 | 88.2 | 57.91 | 98.28 | 84.65 | 97.71 | 52.93 | 55.05 | 68.56 | 96.38 | 59.34 | 45.7 | 67.14 | 93.78 | 56.74 | 75.26 |
| **FT-M** | 99.75 | 99.33 | 54.32 | 93.01 | 86.60 | 100.0 | 62.93 | 45.92 | 69.62 | 100.0 | 74.7 | 54.86 | 76.52 | 100.0 | 90.04 | 95.02 |
| **+ Ours** | 99.75 | 96.8 | 57.08 | 96.54 | 87.54 | 100.0 | 63.91 | 60.4 | 74.77 | 100.0 | 73.62 | 75.34 | 82.99 | 100.0 | 93.46 | 96.73 |
| **LoRA** | 100.0 | 100.0 | 23.34 | 30.44 | 63.45 | 100.0 | 55.41 | 28.29 | 61.23 | 100.0 | 71.92 | 9.99 | 60.64 | 100.0 | 48.84 | 74.42 |
| **+ Ours** | 100.0 | 94.31 | 61.16 | 87.2 | 85.67 | 100.0 | 63.67 | 58.72 | 74.13 | 100.0 | 73.96 | 57.85 | 77.27 | 97.68 | 68.45 | 83.06 |
| **MELO** | 100.0 | 96.77 | 27.11 | 92.35 | 79.06 | 99.13 | 54.04 | 40.96 | 64.71 | 99.0 | 71.78 | 55.83 | 75.54 | 99.97 | 80.77 | 90.37 |
| **+ Ours** | 100.0 | 93.31 | 50.36 | 97.2 | 85.22 | 100.0 | 60.25 | 66.48 | 75.58 | 99.91 | 71.81 | 78.09 | 83.27 | 99.68 | 82.58 | 91.13 |
| **WISE** | 92.42 | 70.86 | 54.57 | 100.0 | 79.46 | - | - | - | - | - | - | - | - | - | - | - |
| **+ Ours** | 97.55 | 76.09 | 54.17 | 100.0 | 81.95 | - | - | - | - | - | - | - | - | - | - | - |
| | **LLaMA 3-8b-Instruct** | | | | | | | | | | | | | | | |
| | Rel. | Gen. | Por. | Loc. | Avg. | Rel. | Por. | Loc. | Avg. | Rel. | Por. | Loc. | Avg. | Rel. | Loc. | Avg. |
| **ROME** | 99.17 | 97.91 | 58.12 | 95.9 | 87.78 | 98.84 | 54.76 | 49.74 | 67.78 | 99.94 | 58.0 | 42.94 | 66.96 | 92.43 | 72.63 | 82.53 |
| **MEMIT** | 96.67 | 92.46 | 58.78 | 98.23 | 86.53 | 98.51 | 53.65 | 48.45 | 66.87 | 99.44 | 57.81 | 42.73 | 66.66 | 96.26 | 71.23 | 83.75 |
| **FT-M** | 100.0 | 99.75 | 40.43 | 79.43 | 79.90 | 100.0 | 57.13 | 30.01 | 62.38 | 100.0 | 72.62 | 31.47 | 68.03 | 100.0 | 92.96 | 96.48 |
| **+ Ours** | 100.0 | 99.75 | 48.63 | 94.78 | 85.79 | 100.0 | 60.88 | 44.67 | 68.52 | 100.0 | 73.5 | 58.29 | 77.26 | 99.99 | 94.87 | 97.43 |
| **LoRA** | 100.0 | 100.0 | 26.55 | 38.85 | 66.35 | 100.0 | 52.99 | 26.46 | 59.82 | 100.0 | 71.1 | 9.02 | 60.04 | 100.0 | 59.77 | 79.88 |
| **+ Ours** | 100.0 | 98.5 | 51.57 | 93.13 | 85.80 | 100.0 | 61.46 | 56.1 | 72.52 | 100.0 | 72.8 | 57.54 | 76.78 | 98.16 | 77.24 | 87.7 |
| **MELO** | 100.0 | 96.84 | 39.63 | 98.8 | 83.82 | 100.0 | 59.07 | 65.78 | 74.95 | 100.0 | 71.55 | 87.77 | 86.44 | 100.0 | 98.56 | 99.28 |
| **+ Ours** | 100.0 | 95.77 | 43.08 | 98.8 | 84.41 | 100.0 | 58.72 | 69.1 | 75.94 | 100.0 | 70.26 | 89.81 | 86.69 | 99.98 | 98.56 | 99.27 |
| **WISE** | 71.67 | 51.29 | 49.27 | 100.0 | 68.06 | - | - | - | - | - | - | - | - | - | - | - |
| **+ Ours** | 82.67 | 62.34 | 47.54 | 100.0 | 73.14 | - | - | - | - | - | - | - | - | - | - | - |

*Table 2.* Editing performance on MQuAKE.

| | LLaMA 2-7b-chat | | | | LLaMA 3-8b-Instruct | | | |
|---|---|---|---|---|---|---|---|---|
| | Rel. | Sng-Hop. | Mlt-Hop. | Avg. | Rel. | Sng-Hop. | Mlt-Hop. | Avg. |
| **FT-M** | 100.0 | 83.0 | 30.0 | 71.0 | 100.0 | 82.0 | 24.0 | 68.67 |
| **+ Ours** | 99.86 | 89.0 | 37.0 | 75.29 | 100.0 | 85.0 | 30.0 | 71.67 |
| **LoRA** | 100.0 | 95.0 | 39.0 | 78.0 | 100.0 | 98.0 | 35.0 | 77.67 |
| **+ Ours** | 99.75 | 93.0 | 48.0 | 80.25 | 100.0 | 95.0 | 40.0 | 78.33 |

ments were even more pronounced when the editing sequence is longer ($T = 10, 100$). Notably, according to our results on ZsRE, LoRA (and FT-M) achieved highly competitive continual editing performance when enhanced with OVERTONE, on par with specialized continual editing methods like MELO and WISE. In contrast, in previous works (Zhang et al., 2024c; Wang et al., 2024d), vanilla LoRA is generally considered unsuitable for continual editing unless significant adaptations are implemented.

To conclude, these results clear demonstrated the flexibility and power of OVERTONE in diverse KE scenarios.

### 4.4. Ablation Studies

We end this section with an ablation study on OVERTONE to showcase how each component contributes to its final performance. Results from editing LLaMA 2 on ZsRE with LoRA are presented in Tab 3. According to the table, we note the

following findings. First, pure token-level smoothing ("w/o clip") increases both portability and locality, confirming that overfitting due to CE loss indeed hurts editing performance. Additionally, the way to smooth target distribution plays a critical role: using the unedited predicted distributions ("w/o dyn-$\pi_{flt}$") leads to significant drop, due to the conflicts raise from the outdated internal knowledge. Extra evidence can be seen from ("w/o chk-$\pi_{flt}$"), where the mixture (Eq (3)) is always applied without checking if the probability of label $y_i$ is the largest. Finally, the noise in predicted distribution $\pi_\theta$ also hinders the editing process: without filtering them out ("w/o flt-$\pi_{flt}$"), both generality and portability decreased. All empirical results aligns well with our analysis in Sec 3.

*Table 3.* Ablation studies on OVERTONE, "w/o clip" sets $\epsilon = 0$, "w/o dyn-$\pi_{flt}$" uses unedited prediction, "w/o chk-$\pi_{flt}$" always adopt the mixture in Eq (3), "w/o flt-$\pi_{flt}$" uses full $\pi_\theta$ without filtering out tail (noisy) regions.

| | LLaMA 2-7b-chat | | | | |
|---|---|---|---|---|---|
| | Rel. | Gen. | Por. | Loc. | Avg. |
| **LoRA** | 100.0 | 100.0 | 23.34 | 30.44 | 63.45 |
| **w/o clip** | 100.0 | 99.75 | 26.6 | 41.08 | 66.86 |
| **w/o dyn-$\pi_{flt}$** | 99.18 | 97.67 | 36.32 | 51.57 | 71.18 |
| **w/o chk-$\pi_{flt}$** | 95.35 | 86.51 | 57.92 | 90.08 | 82.47 |
| **w/o flt-$\pi_{flt}$** | 100.0 | 83.93 | 58.2 | 90.36 | 83.12 |
| **+ Ours** | 100.0 | 94.31 | 61.16 | 87.2 | 85.67 |

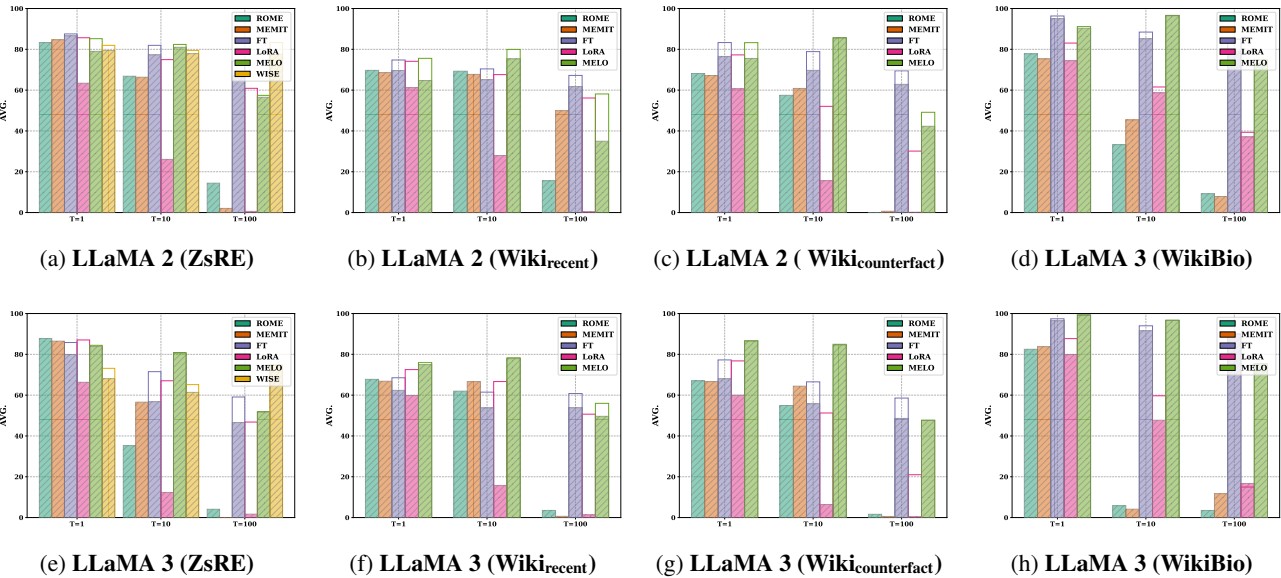

*Figure 3.* Continual Editing performance under different sequence length $T$. Solid and transparent bars show performance with and without OVERTONE. Unfilled area marks the performance gap. ROME and MEMIT didn't use OVERTONE.

## 5. Related Works

Existing KE methods mainly fall into two classes.

**Internal Storage** updates model parameters for the adaptation. Early studies fine-tuned a LLM directly but suffered from severe forgetting problem (Wang et al., 2023b). For more precise editing, Zhu et al. (2020) imposed a relaxed $\ell_2$ norm constraint on parameter updates, and Dong et al. (2022a); Huang et al. (2023) limited the updates to some specific feed-forward network (FFN) layer(s), based on findings that knowledge is often stored therein (Dai et al., 2021). For further refinement, the *locate-and-edit* paradigm (Meng et al., 2022a;b) first identifies the layer storing the knowledge, then modifies its parameters in an analytic form or through least squared solution. On the other hand, PEFT methods such as LoRA- (Hu et al., 2021; Wang et al., 2024c) and ReFT-family (Wu et al., 2024; Liu et al., 2025b) also performed competitive to locating-based solutions (Wu et al., 2023; Zhang et al., 2024c). In general, these works primarily focus on identifying a small set of parameters most relevant to the new knowledge. However, these approaches are typically trained with instance-level loss, overlooking the token-level differences. Therefore, they remain susceptible to HTO in a similar manner and cannot be mitigated by advanced PEFT methods (Chen et al., 2024; Miao et al., 2025). This work addresses HTO, an orthogonal aspect of the KE process, and complements existing studies in a model-agnostic manner. Our OVERTONE is established without assumptions about which parameters are updated, allowing it to be seamlessly integrated with existing methods without compromising their selective nature. We validate our approach by showing that OVERTONE enhances the performance of two representative internal stage methods across diverse scenarios.

**External Storage** resorts to external memories without updating original parameters. This category includes meta-learning-based MEND (Mitchell et al., 2021) and its multi-task varient InstructEdit (Zhang et al., 2024b), in-context learning-based IKE (Zheng et al., 2023), retrieval-based LTE (Jiang et al., 2024), augmentation-based StableKE (Wei et al., 2024), and proxy model-based SERAC (Mitchell et al., 2022). Notwithstanding, these methods often require large-scale, *hard-to-access* dataset for retrieval (e.g., IKE, LTE) as in retrieval-augmented generation (RAG, (Gao et al., 2023; Wang et al., 2024b; Xu et al., 2024; Yu et al., 2025; Liu et al., 2025a; Xu et al., 2025)), or for training auxiliary models (e.g., MEND, InstructEdit, SERAC). As a result, their practicality is limited, and they struggle with Continual Editing that needs frequent updates (Wang et al., 2024d). Recently, specialized methods for Continual Editing have been proposed. These approaches introduce adapters (GRACE (Hartvigsen et al., 2024)), LoRAs (MELO (Yu et al., 2024)), or weight copies (WISE (Wang et al., 2024d)) to memorize new knowledge, and learn gating mechanism to determine whether to use original or new knowledge. The gating mechanisms are often learned through additional representation-distance-based codebooks (Yu et al., 2024) or distinct margin losses (Wang et al., 2024d), making external storage methods more complex. However, like internal storage methods, they optimize editing parameters using instance-level loss functions, ignoring token-level differences. Consequently, they may also suffer from HTO and can benefit from our OVERTONE

framework. Experiments with two external storage methods demonstrate that our solution can be straightforwardly incorporated to more complex KE methods, highlighting the flexibility and versatility of OVERTONE.

**Overfitting and Mitigation** Recent works have identified different forms of KE overfitting and proposed respective mitigation solutions. Namely, Zhang et al. (2024a); Qi et al. (2024) use in-context prompted distribution as the target distribution to fit, which helps improve generalizability (Lampinen et al., 2025), and Ma et al. (2024) focuses on neighboring knowledge perturbation due to the answer-level overfitting. In this work, we focus on understanding and developing generalizable KE. Unlike existing methods, our OVERTONE resorts the model's own prediction to maintain its pretrained knowledge through an adaptive token-level distribution mixing and early stopping, in light of the token-level HTO dynamic.

## 6. Conclusion

We study HTO, a token-dependent overfitting in KE, and show how it degrades an edited LLM's reasoning ability. Inspired by an in-depth analysis on its cause, we propose OVERTONE, which adaptively assigns each token a unique smoothed distribution for better control to mitigate HTO. Our solution enjoys several theoretical advantages, and achieves superior performance on diverse tasks. Encouraged by these promising results, we plan to work on the following directions in our future work. The first direction is to understand how HTO will act on broader KE methods that involves more specialized losses or when facing free-form editing data. The second topic we would like to explore is to unify HTO and other types of KE overfitting, thereby providing a more universal solution. Finally, we advocate for more rigorous experimental design within the KE community—specifically, conducting multiple runs per editing instance—to ensure statistically reliable results.

## Impact Statement

This paper presents work whose goal is to advance the field of Machine Learning. There are many potential societal consequences of our work, none which we feel must be specifically highlighted here.

## Acknowledgment

This work is supported in part by the US National Science Foundation under grant NSF IIS-2141037. Any opinions, findings, and conclusions or recommendations expressed in this material are those of the author(s) and do not necessarily reflect the views of the National Science Foundation.

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

# A. Omitted Theorems and Proofs

In this section we present the full theoretical analysis. All theorems are (re)stated in a formal manner for the convenience of reading.

## A.1. Notations

For completeness we highlight important notations that will be used. Throughout this paper, we use $\mathrm{CE}[\cdot\|\cdot]$ and $\mathrm{D_{KL}}[\cdot\|\cdot]$ to compute cross-entropy and Kullback–Leibler divergence between two distributions respectively. Specifically, given two discrete distributions $p, q$, $\mathrm{CE}[p\|q] = \sum_i -p_i \log q_i$, and $\mathrm{D_{KL}}[p\|q] = \sum_i -p_i \log \frac{q_i}{p_i}$. In addition, $\mathbf{1}(\cdot)$ is the indicator function such that $\mathbf{1}(a) = 1$ if event $a$ holds and 0 otherwise. For $a \in \mathbb{R}^p$, define the $l_2$ norm as $\|a\|_2 = \sqrt{\sum_{i=1}^p a_i^2}$. For $a, b \in \mathbb{R}^p$, define the inner product as $\langle a, b \rangle = a^\top b$. Define the cosine similarity $\cos(a, b) = \frac{\langle a,b \rangle}{\|a\|_2 \|b\|_2}$.

## A.2. OVERTONE is universal and efficient

The first merit of OVERTONE, as stated in the main body, lies in its universality and efficiency.

**Proposition A.1.** *OVERTONE loss generalizes CE loss and reduces to the latter when $\epsilon = 0, \lambda = 1$.*

**Proposition A.2.** *Using Alg 1, the additional computation complexity induced by OVERTONE is $\mathcal{O}(|\mathcal{V}|)$ when fitting a token, where $|\mathcal{V}|$ is the vocabulary size.*

Our proofs rely on the following lemma, which plays a key role in connecting OVERTONE to a regularized loss.

**Lemma A.3.** *Given $y_i$, for an arbitrary token $y$ and context $c$, and $\pi_{tar} = \lambda\delta_{y_i}(y) + \pi_{flt}(y)$, we have*

$$CE[\pi_{tar}(y \mid c)\|\pi_\theta(y \mid c)] = \lambda CE[\delta_{y_i}(y)\|\pi_\theta(y \mid c)] + (1-\lambda)CE[\pi_{flt}(y \mid c) \mid (y \mid c)]. \tag{4}$$

*Proof.* The proof is based on the definition of cross entropy (Cover, 1999).

$$
\begin{aligned}
&\mathrm{CE}[\pi_{\mathrm{tar}}(y \mid c)\|\pi_\theta(y \mid c)] \\
&= -\sum_{i=1}^{|\mathcal{V}|} \pi_{\mathrm{tar}}(y \mid c) \log \pi_\theta(y \mid c) \\
&= -\sum_{i=1}^{|\mathcal{V}|} (\lambda\delta_{y_i}(y) + (1-\lambda)\pi_{\mathrm{flt}}(y \mid c)) \log \pi_\theta(y \mid c) \\
&= -\left( \lambda\sum_{i=1}^{|\mathcal{V}|} \delta_{y_i}(y) \log \pi_\theta(y \mid c) + (1-\lambda)\sum_{i=1}^{|\mathcal{V}|} \pi_{\mathrm{flt}}(y \mid c) \log \pi_\theta(y \mid c) \right) \\
&= \lambda\mathrm{CE}[\delta_{y_i}(y)\|\pi_\theta(y \mid c)] + (1-\lambda)\mathrm{CE}[\pi_{\mathrm{flt}}(y \mid c)\|\pi_\theta(y \mid c)]. \tag{5}
\end{aligned}
$$

This completes our proof. □

We are ready to prove Prop 3.1.

*Proof.* The proof is based on the fact that OVERTONE objective minimizes a *forward* KL-divergence, which is equivalent to

minimizing cross-entropy (Cover, 1999; Bishop & Nasrabadi, 2006). Namely,

$$
\begin{aligned}
\ell_{O\text{VER}T\text{ONE}}(\theta) &\triangleq \sum_{j=1}^{m} \max(\mathrm{D}_{\mathrm{KL}}[\pi_{\mathrm{tar}}(y \mid \boldsymbol{c}_i) \| \pi_\theta(y \mid \boldsymbol{c}_i)], \epsilon) \\
&= \sum_{j=1}^{m} \mathrm{D}_{\mathrm{KL}}[\pi_{\mathrm{tar}}(y \mid \boldsymbol{c}_i) \| \pi_\theta(y \mid \boldsymbol{c}_i)] \mathbf{1}\left(\mathrm{D}_{\mathrm{KL}}[\pi_{\mathrm{tar}}(y \mid \boldsymbol{c}_i) \| \pi_\theta(y \mid \boldsymbol{c}_i)] > \epsilon\right) \\
&\stackrel{(a)}{=} \sum_{j=1}^{m} \left(\mathrm{CE}[\pi_{\mathrm{tar}}^{(j)} \| \pi_\theta^{(j)}] + H(\pi_{\mathrm{tar}}^{(j)})\right) \mathbf{1}\left(\mathrm{D}_{\mathrm{KL}}[\pi_{\mathrm{tar}}^{(j)}] \| \pi_\theta^{(j)}] > \epsilon\right) \\
&= \sum_{j=1}^{m} \mathrm{CE}[\pi_{\mathrm{tar}}^{(j)} \| \pi_\theta^{(j)}] \mathbf{1}\left(\mathrm{D}_{\mathrm{KL}}[\pi_{\mathrm{tar}}^{(j)}] \| \pi_\theta^{(j)}] > \epsilon\right) + C.
\end{aligned}
\tag{6}
$$

Starting from step $(a)$, we denote $\pi_{\mathrm{tar}}^{(j)} = \pi_{\mathrm{tar}}(y \mid \boldsymbol{c}_i)$ and $\pi_\theta^{(j)}$ similarly for brevity, $C$ denotes terms that are constant to learnable parameter $\theta$. Therefore, setting $\epsilon = 0$ gets us rid of the indicator term. Further plug in Eq (5), we see that setting $\lambda = 1$ reduces to the standard CE loss. This completes the proof.

$\square$

In terms of Prop 3.2, the computation overhead can be seen by checking Algo 1.

*Proof.* The additional computation complexity of OVERTONE is due to line 8-10 in Algo 1. These steps involve finding the maximal logits, pruning small logits, and compute the probability with softmax function from the pruned logits. All of them have linear time complexity $|\mathcal{V}|$. This completes our proof.

$\square$

## A.3. OVERTONE provides better updates

We present the formal analysis of how OVERTONE provides better parameters update as outlined in Thm 3.3. Our analysis is established in the same spirit of influence function (Koh & Liang, 2017).

We first restate Thm 3.3, which outlines the two aspects where OVERTONE is better than training standard CE loss.

**Theorem A.4** (Informal). *Under regularity conditions, compared to optimizing the vanilla CE loss, OVERTONE provides a more favorable update direction for the parameters and has less influence on unrelated knowledge.*

The formal statement is as follows.

**Theorem A.5** (Formal). *Let $G$ be the ideal gradient of retraining the LLM using $\hat{\theta}^{old}$ as the initial value, as defined in Eq (8). Considering the simplified case where $\epsilon = 0$ in Eq (6), under Assumptions A.6 and A.7, there exists some $\lambda \in [0, 1]$ such that*

$$
\cos\left(\nabla_\theta \ell_{CE}(z^{new}; \hat{\theta}^{old}), G\right) < \cos\left(\nabla_\theta \ell_O\text{VERTONE}(z^{new}; \hat{\theta}^{old}), G\right).
$$

*In other words, using the OVERTONE loss provides a better approximation of the direction of $G$ compared to the standard CE loss, meaning the gradient direction is closer to $G$.*

*Now, denote the new estimator obtained through either $\ell_{CE}$ or $\ell_O$VERTONE by $\hat{\theta}_{CE}^{new}$ or $\hat{\theta}_O^{new}$VERTONE, respectively. Let $Z^{un} = (X^{un}, Y^{un})$ be a random vector representing unrelated data. Under Assumptions A.11 and A.13, we have*

$$
\mathbb{E}_{Z^{un}}\left[\left|\pi_{\hat{\theta}_O^{new}\text{VERTONE}}(Z^{un}) - \pi_{\hat{\theta}^{old}}(Z^{un})\right|\right] < \mathbb{E}_{Z^{un}}\left[\left|\pi_{\hat{\theta}_{CE}^{new}}(Z^{un}) - \pi_{\hat{\theta}^{old}}(Z^{un})\right|\right].
$$

*This result indicates that updates based on the OVERTONE loss induce smaller deviations in the predicted distribution for unrelated data compared to updates based on the standard CE loss, thereby better preserving locality.*

Theorem A.5 consists of two parts: Theorem A.10 and Theorem A.15. Theorem A.10 states that our method provides a more effective direction for parameter updates, while Theorem A.15 asserts that our method results in a smaller perturbation on unrelated knowledge. The assumptions and proofs will be presented in Sections A.3.1 and A.3.2, respectively.

A.3.1. OUR METHOD GIVES A BETTER DIRECTION OF PARAMETER UPDATES

Without loss of generality, suppose that a LLM is pretrained on some large textual corpus $\{z_n\}_{n=1}^N$, each training sample $z_n = (\boldsymbol{x}_n, \boldsymbol{y}_n)$ where $\boldsymbol{y}_n = (y_1, \cdots, y_{m_n})$. KE involves updating some knowledge carried by $z^{\text{old}} = (\boldsymbol{x}, \boldsymbol{y}^{\text{old}})$ to new $z^{\text{new}} = (\boldsymbol{x}, \boldsymbol{y}^{\text{new}})$. Let $\hat{\theta}^{\text{old}}$ denote the pre-trained LLM parameters. Given this piece of new knowledge, the ideal LLM should have parameters $\hat{\theta}^{\text{new}}$ from a full retraining by solving

$$\min_\theta \frac{1}{N} \sum_{n=1}^N \ell_{\text{CE}}(z_n; \theta) - \frac{1}{N} \ell_{\text{CE}}(z^{\text{old}}; \theta) + \frac{1}{N} \ell_{\text{CE}}(z^{\text{new}}; \theta), \tag{7}$$

where $\ell_{\text{CE}}$ denotes the standard CE loss. In general, we define $\ell_\delta(\theta)$ as

$$\ell_\delta(\theta) = \sum_{i=1}^n \ell_{\text{CE}}(z_i; \theta) + \delta\big(\ell_{\text{CE}}(z^{\text{new}}; \theta) - \ell_{\text{CE}}(z^{\text{old}}; \theta)\big).$$

Moreover define

$$\hat{\theta}_\delta = \arg\min_\theta \ell_\delta(\theta).$$

So we find that $\hat{\theta}_0 = \hat{\theta}^{\text{old}}$ and $\hat{\theta}_{\frac{1}{N}} = \hat{\theta}^{\text{new}}$. Starting from $\hat{\theta}^{\text{old}}$, when we perform gradient descent by using loss $\ell_{\frac{1}{N}}(\theta)$ to retrain the model, the gradient will be

$$G \triangleq \nabla_\theta \ell_{\frac{1}{N}}(\hat{\theta}^{\text{old}}). \tag{8}$$

So we just take $G$ as the *optimal* direction to represent that if we retrained the LLM, i.e., the direction of the gradient descent at $\hat{\theta}^{\text{old}}$.

We make following assumption on $\hat{\theta}^{\text{old}}$ such that it is a local the minimizer of $\ell_0(\theta)$.

**Assumption A.6.** The pretrained LLM is converged, namely, $\nabla_\theta \ell_0(\hat{\theta}^{\text{old}}) = 0$.

For brevity, denote

$$\begin{aligned}
a &= \nabla_\theta \ell_{\text{CE}}(z^{\text{new}}; \hat{\theta}^{\text{old}}), \\
b &= -\nabla_\theta \ell_{\text{CE}}(z^{\text{old}}; \hat{\theta}^{\text{old}}), \\
c &= \sum_{i=1}^m \nabla_\theta \text{CE}[\pi_{\text{flt}}(y \mid \boldsymbol{c}_i^{\text{new}}) \| \pi_\theta(y \mid \boldsymbol{c}_i^{\text{new}})]\Big|_{\theta=\hat{\theta}^{\text{old}}}.
\end{aligned} \tag{9}$$

**Assumption A.7.** $\cos(b, c)$ satisfies

$$\cos(b, c) > 1 - \frac{\|b\|_2^2}{8\|a+b\|_2^2}(1 - \cos(a, a+b))^2. \tag{10}$$

*Remark* A.8 (Interpretation of the Assumption A.7). The Assumption A.7 ensure direction $b$ and $c$ will not be far away. Roughly speaking, when we take $\frac{\|b\|_2^2}{8\|a+b\|_2^2}$ as some constant. It says that $1 - \cos(b, c) < (1 - \cos(a, a+b))^2$, which means the directions of $b$ and $c$ are closer compared with $a$ and $a + b$. When we look it more carefully, Note that $a$ represents $\nabla_\theta \ell_{\text{CE}}(z^{\text{new}}; \hat{\theta}^{\text{old}})$ and $a + b$ represents the ideal direction $G$. Since the old knowledge gradient $b$ is present, directly fine-tuning $\ell_{\text{CE}}$ (i.e., the baseline method) results in a deviation compared with the ideal direction $G$. This directional deviation is measured by $\cos(a, a+b)$. Let $S^{(i)}$ denote the collection of *unfiltered* tokens in $\pi_{\text{flt}}(y \mid \boldsymbol{c}_i^{\text{new}})$,

$$b = -\nabla_\theta \ell_{\text{CE}}(z^{\text{old}}; \hat{\theta}^{\text{old}}) = \sum_{i=1}^m \nabla_\theta \log \pi_\theta(y_i^{\text{old}} \mid \boldsymbol{c}_i^{\text{old}})\Big|_{\theta=\hat{\theta}^{\text{old}}}, \tag{11}$$

$$c = \sum_{i=1}^m \nabla_\theta \text{CE}[\pi_{\text{flt}}(y \mid \boldsymbol{c}_j^{\text{new}}) \| \pi_\theta(y \mid \boldsymbol{c}_j^{\text{new}})]\Big|_{\theta=\hat{\theta}^{\text{old}}} = -\sum_{i=1}^m \sum_{y \in S^{(i)}} \pi_{\text{flt}}(y \mid \boldsymbol{c}_i^{\text{new}}) \nabla_\theta \log \pi_\theta(y \mid \boldsymbol{c}_i^{\text{new}})\Big|_{\theta=\hat{\theta}^{\text{old}}}. \tag{12}$$

Given the new knowledge $\boldsymbol{c}_j^{\text{new}}$, when $y \in S^{(i)}$, it implies that $y$ is likely close to $y_i^{\text{old}}$ with some probability. Compared to the scenario where the old knowledge $\boldsymbol{c}_i^{\text{old}}$ is given, the gradients $\nabla_\theta \log \pi_\theta(y_i^{\text{old}} \mid \boldsymbol{c}_i^{\text{old}})\big|_{\theta=\hat{\theta}^{\text{old}}}$ and $\nabla_\theta \log \pi_\theta(y \mid \boldsymbol{c}_i^{\text{new}})\big|_{\theta=\hat{\theta}^{\text{old}}}$

tend to point in opposite directions. This is because both gradients are evaluated at $y^{\text{old}}$ or a point close to $y^{\text{old}}$, but the first is conditioned on $c_j^{\text{old}}$ while the second is conditioned on $c_j^{\text{new}}$. Equivalently, this implies that $b$ and $c$ are aligned in the same direction. To ensure that we can find a closer direction, we require $b$ and $c$ to be approximately as close as $a$ and $a + b$. Our goal is to align with the negative gradient direction of the old knowledge. This ensures that when leveraging the information from $c$ to weight our method, we can identify a direction that closely approximates the ideal optimization direction.

*Remark* A.9. To elaborate further, we take logistic regression as an example for illustration.

When considering only the $k$-th token, for a training point $z_k = (c_k, y_k)$, let $p(y_k \mid c_k) = \sigma(y_k\theta^\top c_k)$, where $y_k \in \{-1, 1\}$ and $\sigma(t) = \frac{1}{1+\exp(-t)}$ is the sigmoid function. the gradient of the log-probability with respect to $\theta$ is given by:

$$\nabla_\theta \log p(z_k, \theta) = \sigma(-y_k\theta^\top c_k)y_k c_k.$$

Then, we find that:

$$b = \sigma(-y_k^{\text{old}}\theta^\top c_k^{\text{old}})y_k^{\text{old}} c_k^{\text{old}},$$

$$c = -\sum_{y_k \in S^{(i)}} p_{y_k}\sigma(-y_k\theta^\top c_k^{\text{new}})y_k c_k^{\text{new}} = -p_{\text{old}}\sigma(-y_k^{\text{old}}\theta^\top c_k^{\text{new}})y_k^{\text{old}} c_k^{\text{new}} - p_{\text{new}}\sigma(-y_k^{\text{new}}\theta^\top c_k^{\text{new}})y_k^{\text{new}} c_k^{\text{new}}.$$

This follows from the fact that $y_k \in \{-1, 1\}$. Note that $c_k^{\text{new}}$ and $c_k^{\text{old}}$ may be far apart, and $p_{\text{old}}$ is likely to be large since $\pi_{\text{flt}}$ is a denoised version of $\pi_\theta$, meaning it contains less noise (Tang et al., 2024). As a result, the directions of $b$ and $c$ will be close.

**Theorem A.10.** *Let $G$ be the ideal gradient of retraining the LLM using $\hat{\theta}^{\text{old}}$ as the initial value, as defined in Eq (8). Considering the simplified case where $\epsilon = 0$ in Eq (6), under Assumptions A.6 and A.7, there exists some $\lambda \in [0, 1]$ such that*

$$\cos\left(\nabla_\theta\ell_{CE}(z^{new}; \hat{\theta}^{old}), G\right) < \cos\left(\nabla_\theta\ell_O\text{VERTONE}(z^{new}; \hat{\theta}^{old}), G\right).$$

*In other words, using the OVERTONE loss provides a better approximation of the direction of $G$ compared to the standard CE loss, in the sense that OVERTONE gradient direction is closer to $G$.*

*Proof.* First, by definition, the optimal gradient direction $G$ when using $\theta^{\text{old}}$ as the initial value is given by

$$\begin{aligned}
G &= \nabla_\theta\ell_{\frac{1}{N}}(\hat{\theta}^{\text{old}}) \\
&= \nabla_\theta\ell_0(\hat{\theta}^{\text{old}}) + \frac{1}{N}\left(\nabla_\theta\ell_{\text{CE}}(z^{\text{new}}; \hat{\theta}^{\text{old}}) - \nabla_\theta\ell_{\text{CE}}(z^{\text{old}}; \hat{\theta}^{\text{old}})\right) \\
&\overset{(a)}{=} \frac{1}{N}\left(\nabla_\theta\ell_{\text{CE}}(z^{\text{new}}; \hat{\theta}^{\text{old}}) - \nabla_\theta\ell_{\text{CE}}(z^{\text{old}}; \hat{\theta}^{\text{old}})\right),
\end{aligned}$$

where $(a)$ holds from the stationary condition of $\hat{\theta}^{\text{old}}$ as per Assumption A.6. Note that this optimal direction is inaccessible since it is infeasible to find the ground truth $z^{\text{old}}$ wherefrom the LLM's old knowledge is learned. In practice, only $z^{\text{new}}$ is available, which is provided by the user.

To see that OVERTONE can provide a better direction, we check the gradient of CE loss $\ell_{\text{CE}}$ and our loss $\ell_{OVERTONE}$. Recall the definition of $a, b, c$ given by Eq (9), for CE loss, we have

$$\nabla_\theta\ell_{\text{CE}}(z^{\text{new}}; \theta) = -\sum_{i=1}^m \nabla_\theta \log \pi_\theta(y_i^{\text{new}} \mid c_i^{\text{new}}) = a, \tag{13}$$

where $c_i^{\text{new}} = x \oplus y_{<i}^{\text{new}}$, as derived in Sec 3 in the main body.

For OVERTONE loss, according to Eq (5) and Eq (6), we have

$$\nabla_\theta \ell_O \text{VERTONE}(z^{\text{new}}; \theta) = \sum_{i=1}^m \nabla_\theta \text{CE}[\pi_{\text{tar}}(y \mid \boldsymbol{c}_i^{\text{new}}) \| \pi_\theta(y \mid \boldsymbol{c}_i^{\text{new}})]$$

$$= \lambda \sum_{i=1}^m \nabla_\theta \text{CE}\big[\delta_{y_i^{\text{new}}}(y) \| \pi_\theta(y \mid \boldsymbol{c}_i^{\text{new}})\big] + (1-\lambda) \sum_{i=1}^m \nabla_\theta \text{CE}[\pi_{\text{flt}}(y \mid \boldsymbol{c}_i^{\text{new}}) \| \pi_\theta(y \mid \boldsymbol{c}_i^{\text{new}})]$$

$$= -\left( \lambda \sum_{i=1}^m \nabla_\theta \log \pi_\theta(y_i \mid \boldsymbol{c}_i^{\text{new}}) + (1-\lambda) \sum_{i=1}^m -\nabla_\theta \text{CE}[\pi_{\text{flt}}(y \mid \boldsymbol{c}_i^{\text{new}}) \| \pi_\theta(y \mid \boldsymbol{c}_i^{\text{new}})] \right)$$

$$= \lambda a + (1-\lambda)c.$$

Next, we check cosine similarity $\cos\left(\nabla_\theta \ell_{\text{CE}}(z^{\text{new}}; \hat{\theta}^{\text{old}}), G\right)$ and $\cos\left(\nabla_\theta \ell_O \text{VERTONE}(z^{\text{new}}; \hat{\theta}^{\text{old}}), G\right)$. A larger cosine similarity indicates an update direction that aligns with the ideal $G$ better and is more effective.

Note that

$$\cos\left(\nabla_\theta \ell_{\text{CE}}(z^{\text{new}}; \hat{\theta}^{\text{old}}), G\right) = \frac{\langle a, a+b \rangle}{\|a\|_2 \|(a+b)\|_2},$$

$$\cos\left(\nabla_\theta \ell_O \text{VERTONE}(z^{\text{new}}; \hat{\theta}^{\text{old}}), G\right) = \frac{\langle \lambda a + (1-\lambda)c, a+b \rangle}{\|\lambda a + (1-\lambda)c\|_2 \|(a+b)\|_2}.$$

We will show that, there $\exists \lambda \in [0, 1]$, s.t.

$$\frac{\langle a, a+b \rangle}{\|a\|_2} < \frac{\langle \lambda a + (1-\lambda)c, a+b \rangle}{\|\lambda a + (1-\lambda)c\|_2}.$$

We further denote $\delta_{bc} = \frac{c}{\|c\|_2} - \frac{b}{\|b\|_2}$ which quantifies the directional difference between $b$ and $c$. We then have:

$$c = \left( \frac{b}{\|b\|_2} + \delta_{bc} \right) \|c\|_2. \tag{14}$$

Take $\lambda = \frac{\|c\|_2}{\|b\|_2 + \|c\|_2}$, by substituting $c$ by Eq (14) and applying the triangle inequality, we obtain

$$\frac{\langle \lambda a + (1-\lambda)c, a+b \rangle}{\|\lambda a + (1-\lambda)c\|_2} = \frac{\left\langle \left(\frac{\|c\|_2}{\|b\|_2+\|c\|_2}\right)a + \left(\frac{\|b\|_2\|c\|_2}{\|b\|_2+\|c\|_2}\right)\left(\frac{b}{\|b\|_2} + \delta_{bc}\right), a+b \right\rangle}{\left\| \left(\frac{\|c\|_2}{\|b\|_2+\|c\|_2}\right)(a+b) + \left(\frac{\|b\|_2\|c\|_2}{\|b\|_2+\|c\|_2}\right)\delta_{bc} \right\|_2}$$

$$\geq \frac{\|a+b\|_2^2 - \|a+b\|_2(\|\delta_{bc}\|_2\|b\|_2)}{\|a+b\|_2 + \|b\|_2\|\delta_{bc}\|_2}$$

$$\geq \frac{\|a+b\|_2 - \|b\|_2\|\delta_{bc}\|_2}{\|a+b\|_2 + \|b\|_2\|\delta_{bc}\|_2}\|a+b\|_2.$$

Therefore, to show OVERTONE provides a larger cosine similarity, it suffices to show that

$$\frac{\|a+b\|_2 - \|b\|_2\|\delta_{bc}\|_2}{\|a+b\|_2 + \|b\|_2\|\delta_{bc}\|_2} > \cos(a, a+b),$$

which is equivalent to show

$$\|\delta_{bc}\|_2 < \frac{\|b\|_2}{\|a+b\|_2}\left(\frac{1 - \cos(a, a+b)}{1 + \cos(a, a+b)}\right).$$

Note that $\|\delta_{bc}\|_2^2 = 2 - 2\cos(b, c)$, it suffices to show

$$\cos(b, c) > 1 - \frac{\|b\|_2^2}{2\|a+b\|_2^2}\left(\frac{1 - \cos(a, a+b)}{1 + \cos(a, a+b)}\right)^2.$$

Since $\cos(a, a+b) \leq 1$, this condition holds from Assumption A.7. This completes our proof. $\qquad\square$

A.3.2. OUR METHOD LEADS TO A SMALLER PERTURBATION ON UNRELATED KNOWLEDGE.

Now denote our new estimator obatined through either $\ell_{\text{CE}}$ or $\ell_O$ VERTONE by $\hat{\theta}_{\text{CE}}^{\text{new}}$ or $\hat{\theta}_O^{\text{new}}$ VERTONE. After updating the model parameters to incorporate new knowledge, it is crucial to assess whether this update introduces significant changes to unrelated data.

Without loss of generality, let $z^{\text{un}} = (x^{\text{un}}, y^{\text{un}})$ represent a query-answer pair, where $x^{\text{un}}$ is an unrelated query and $y^{\text{un}}$ is its corresponding predicted answer. To ensure good *locality*, the predicted distribution on $z^{\text{un}}$ should remain unchanged against modifications introduced by the update, ensuring that the model's behavior on unaffected regions of the data distribution is preserved. That means we want to compare $\left| \pi_{\hat{\theta}_{\text{CE}}^{\text{new}}}(z^{\text{un}}) - \pi_{\hat{\theta}^{\text{old}}}(z^{\text{un}}) \right|$ with $\left| \pi_{\hat{\theta}_O^{\text{new}} \text{VERTONE}}(z^{\text{un}}) - \pi_{\hat{\theta}^{\text{old}}}(z^{\text{un}}) \right|$.

Now, treating $Z^{\text{un}} = (X^{\text{un}}, Y^{\text{un}})$ as a random vector following a certain distribution, we define

$$W \triangleq \nabla_\theta \pi_\theta(Z^{\text{un}}) \Big|_{\theta = \hat{\theta}^{\text{old}}}.$$

Since $W$ is a function of $Z^{\text{un}}$, it is also a random vector. In particular, we introduce the following assumption.

**Assumption A.11.** Assume that $\frac{W}{\|W\|_2}$ and $\|W\|_2$ are independent. Furthermore, assume that

$$\frac{W}{\|W\|_2} \sim \mathcal{U}(\mathbb{S}^{d-1}),$$

where $\mathcal{U}(\mathbb{S}^{d-1})$ denotes the uniform distribution on the unit sphere in $\mathbb{R}^d$ with $d$ denoting the dimensionality of the parameter space.

*Remark* A.12. Since it represents the gradient of the loss evaluated on unrelated data, we lack any prior information about $W$. Given that, we assume that $\frac{W}{\|W\|_2}$ is isotropically distributed.

Recall the definition of $a, b, c$ given by Eq (9), we define $\kappa_R = \frac{\|c\|_2}{\|a\|_2}$.

**Assumption A.13.** We assume that $\kappa_R < 1$.

*Remark* A.14 (Interpretation of the Assumption A.13). As shown in Eq. (12) and Eq. (13):

$$a = -\sum_{i=1}^m \nabla_\theta \log \pi_\theta(y_i^{\text{new}} \mid c_i^{\text{new}}),$$

$$c = -\sum_{i=1}^m \sum_{y \in S^{(i)}} \pi_{\text{flt}}(y \mid c_i^{\text{new}}) \nabla_\theta \log \pi_\theta(y \mid c_i^{\text{new}}) \Big|_{\theta = \hat{\theta}^{\text{old}}}.$$

This implies that $c$ is a weighted combination of $a$ and contributions from other values of $y \in S^{(i)}$. Note that at $\hat{\theta}^{\text{old}}$, given $c_i^{\text{new}}$, when $y \neq y_i^{\text{new}}$, the other points are closer to $y_i^{\text{old}}$. Since the loss has already reached its minimum, these other points tend to have smaller gradient norms compared to $y_i^{\text{new}}$.

**Theorem A.15.** *Let $Z^{un} = (X^{un}, Y^{un})$ be a random vector representing unrelated data. Under Assumptions A.11 and A.13, we have*

$$\mathbb{E}_{Z^{un}}\left[ \left| \pi_{\hat{\theta}_O^{new} \text{VERTONE}}(Z^{un}) - \pi_{\hat{\theta}^{old}}(Z^{un}) \right| \right] < \mathbb{E}_{Z^{un}}\left[ \left| \pi_{\hat{\theta}_{CE}^{new}}(Z^{un}) - \pi_{\hat{\theta}^{old}}(Z^{un}) \right| \right].$$

*This result indicates that updates based on the OVERTONE loss induce smaller deviations in the predicted distribution for unrelated data compared to updates based on the standard CE loss, thereby better preserving locality.*

*Proof.* Again let $\hat{\theta}^{\text{old}}$ denote the pretrained parameters. For any new parameters $\tilde{\theta}^{\text{new}}$, the change of $\pi_\theta(z^{\text{un}})$ when $\theta$ moves from $\hat{\theta}^{\text{old}}$ to $\tilde{\theta}^{\text{new}}$ can be approximated by the first-order Taylor expansion with

$$\pi_{\tilde{\theta}^{\text{new}}}(z^{\text{un}}) - \pi_{\hat{\theta}^{\text{old}}}(z^{\text{un}}) = \nabla_\theta \pi_\theta(z^{\text{un}}) \Big|_{\theta = \hat{\theta}^{\text{old}}}^\top \left( \tilde{\theta}^{\text{new}} - \hat{\theta}^{\text{old}} \right) + o\left( \left\| \tilde{\theta}^{\text{new}} - \hat{\theta}^{\text{old}} \right\|_2 \right).$$

Note that when we perform one step gradient descent, the parameter change can further be expressed by

$$\tilde{\theta}^{\text{new}} - \hat{\theta}^{\text{old}} = -\alpha \nabla_\theta \ell(z^{\text{new}}; \hat{\theta}^{\text{old}}),$$

where $\ell(z^{\text{new}}; \theta)$ can be either CE loss or OVERTONE loss, and $\alpha$ denotes the learning rate.

Then to show OVERTONE leads to smaller perturbation in expectation, it suffices to show that there exists $\lambda \in [0, 1]$ such that

$$\mathbb{E}\big[\big|a^\top W\big|\big] > \mathbb{E}\big[\big|\lambda a^\top W + (1 - \lambda)c^\top W\big|\big].$$

By triangle inequality, we only need to show

$$\mathbb{E}\big[\big|a^\top W\big|\big] > \mathbb{E}\big[\big|c^\top W\big|\big].$$

Finally, by Assumption A.11, $\frac{W}{\|W\|_2} \sim \mathcal{U}(\mathbb{S}^{d-1})$ and $\frac{W}{\|W\|_2}$ and $\|W\|_2$ are independent, we have

$$\frac{\mathbb{E}\Big[\Big|c^\top \frac{W}{\|W\|_2}\Big|\|W\|_2\Big]}{\mathbb{E}\Big[\Big|a^\top \frac{W}{\|W\|_2}\Big|\|W\|_2\Big]} = \frac{\mathbb{E}\Big[\Big|c^\top \frac{W}{\|W\|_2}\Big|\Big]\mathbb{E}[\|W\|_2]}{\mathbb{E}\Big[\Big|a^\top \frac{W}{\|W\|_2}\Big|\Big]\mathbb{E}[\|W\|_2]} = \kappa_R < 1.$$

This completes our proof. □

### A.4. Connection between OVERTONE and DPO

We end up this section by the following analysis on the connection between OVERTONE and direct preference optimization (Rafailov et al., 2024).

**Theorem A.16.** *Let $\epsilon = 0$, then optimizing OVERTONE directly can be seen as optimizing an unbiased estimate of a DPO objective plus some additional KL penalty.*

*Proof.* From Prop 3.1 and Lem A.3, at step $i$, we have the negative loss (objective) to maximize

$$\begin{aligned}
-\ell_{\text{OVERTONE},i}(\theta) &= -D_{\text{KL}}[\pi_{\text{tar}}(y \mid \boldsymbol{c}_i)\|\pi_\theta(y \mid \boldsymbol{c}_i)] \\
&= -\left(\lambda\text{CE}[\delta_{y_i}(y)\|\pi_\theta(y \mid \boldsymbol{c}_i)] + (1 - \lambda)\text{CE}[\pi_{\text{flt}}(y \mid \boldsymbol{c}_i)\|\pi_\theta(y \mid \boldsymbol{c}_i)]\right) \\
&= -\lambda\left(\text{CE}[\delta_{y_i}(y)\|\pi_\theta(y \mid \boldsymbol{c}_i)] - \text{CE}[\pi_{\text{flt}}(y \mid \boldsymbol{c}_i)\|\pi_\theta(y \mid \boldsymbol{c}_i)]\right) - \text{CE}[\pi_{\text{flt}}(y \mid \boldsymbol{c}_i)\|\pi_\theta(y \mid \boldsymbol{c}_i)] \qquad (15) \\
&= \lambda\left(\log \pi_\theta(y_i \mid \boldsymbol{c}_i) + \text{CE}[\pi_{\text{flt}}(y \mid \boldsymbol{c})\|\pi_\theta(y \mid \boldsymbol{c}_i)]\right) - \text{CE}[\pi_{\text{flt}}(y \mid \boldsymbol{c}_i)\|\pi_\theta(y \mid \boldsymbol{c}_i)] \qquad (16)
\end{aligned}$$

From the lens of DPO, note that the editing knowledge $(\boldsymbol{x}, \boldsymbol{y})$ can be seen as a *preferred sample* drawn from unknown $\pi^+$ (e.g., retraining the LM from scratch). Consequently, Eq (16) is in fact an *unbiased estimator* of

$$\lambda\left(\underbrace{\mathbb{E}_{y^+ \sim \pi^+(y|\boldsymbol{c}_i)}[\log \pi_\theta(y^+ \mid \boldsymbol{c}_i)]}_{\text{Preferred distriibution}} - \mathbb{E}_{y^- \sim \pi_{\text{flt}}(y|\boldsymbol{c}_i)}[\log \pi_\theta(y^- \mid \boldsymbol{c}_i)]\right) - \text{CE}[\pi_{\text{flt}}(y \mid \boldsymbol{c}_i)\|\pi_\theta(y \mid \boldsymbol{c}_i)]$$

$$= \lambda\mathbb{E}_{y^+, y^-}\left[\log \frac{\pi_\theta(y^+ \mid \boldsymbol{c}_i)}{\pi_\theta(y^- \mid \boldsymbol{c}_i)}\right] - D_{\text{KL}}[\pi_{\text{flt}}(y \mid \boldsymbol{c}_i)\|\pi_\theta(y \mid \boldsymbol{c}_i)] + C$$

$$\overset{(a)}{=} \lambda\left(\mathbb{E}_{y^+, y^-}\left[\log \frac{\pi_\theta(y^+ \mid \boldsymbol{c}_i)}{\pi_\theta(y^- \mid \boldsymbol{c}_i)} - \log \frac{\pi_{\text{flt}}(y^+ \mid \boldsymbol{c}_i)}{\pi_{\text{flt}}(y^- \mid \boldsymbol{c}_i)}\right] + \mathbb{E}_{y^+}[\log \pi_{\text{flt}}(y^+ \mid \boldsymbol{c}_i) - \mathbb{E}_{y^-}[\log \pi_{\text{flt}}(y^- \mid \boldsymbol{c}_i)]\right)$$
$$- D_{\text{KL}}[\pi_{\text{flt}}(y \mid \boldsymbol{c}_i)\|\pi_\theta(y \mid \boldsymbol{c}_i)] + C$$

$$= \lambda\left(\mathbb{E}_{y^+, y^-}\left[\log \frac{\pi_\theta(y^+ \mid \boldsymbol{c}_i)}{\pi_\theta(y^- \mid \boldsymbol{c}_i)} - \log \frac{\pi_{\text{flt}}(y^+ \mid \boldsymbol{c}_i)}{\log \pi_{\text{flt}}(y^- \mid \boldsymbol{c}_i)}\right] + \underbrace{\mathbb{E}_{y^+}[\pi_{\text{flt}}(y^+ \mid \boldsymbol{c}_i)] - \mathbb{E}_{y^-}[\pi_{\text{flt}}(y^- \mid \boldsymbol{c}_i)]}_{\text{constant wrt } \theta}\right)$$
$$- D_{\text{KL}}[\pi_{\text{flt}}(y \mid \boldsymbol{c}_i)\|\pi_\theta(y \mid \boldsymbol{c}_i)] + C$$

$$= \underbrace{\mathbb{E}_{y^+, y^-}\left[\lambda\log \frac{\pi_\theta(y^+ \mid \boldsymbol{c}_i)}{\pi_{\text{flt}}(y^+ \mid \boldsymbol{c}_i)} - \lambda\log \frac{\pi_\theta(y^- \mid \boldsymbol{c}_i)}{\pi_{\text{flt}}(y^- \mid \boldsymbol{c}_i)}\right]}_{\text{DPO with a clipped exponential preference}} - \underbrace{D_{\text{KL}}[\pi_{\text{flt}}(y \mid \boldsymbol{c}_i)\|\pi_\theta(y \mid \boldsymbol{c}_i)]}_{\text{Additional Penalty}} + C, \qquad (17)$$

where the first term incorporates a *preferred* distribution, of which the user-provided new knowledge $y_i$ serves an unbiased estimate. Step (a) plugs in the log-likelihood ratio between the $(y^+, y^-)$ pair from $\pi_{\text{flt}}$, which is constant with respect to $\theta$

and doesn't affect the objective thereof. In the final step, we treat the first term as a token-level DPO objective using current $\pi_{\text{flt}}$ as the *reference* model, and the preference model is given by a *clipped exponential preference model*

$$\Pr(y^+ \succ y^- \mid \boldsymbol{c}_i) = \min(\exp\big(r(\boldsymbol{c}_i, y^+) - r(\boldsymbol{c}_i, y^-)\big)/Z, 1),$$

where $Z \geq 1$ is some constant. Notably, since our base distribution, $\pi_{\text{flt}}$, is the clipped version of $\pi_\theta$, and $\lambda \in [0, 1]$, the difference in probability of $y^+(y^-)$ given $\boldsymbol{c}_i$ is expected small, so that we can impose

$$0 \leq \lambda \log \frac{\pi_\theta(y^+ \mid \boldsymbol{c}_i)}{\pi_{\text{flt}}(y^+ \mid \boldsymbol{c}_i)} - \lambda \log \frac{\pi_\theta(y^- \mid \boldsymbol{c}_i)}{\pi_{\text{flt}}(y^- \mid \boldsymbol{c}_i)} \leq 1,$$

this allows us to set $Z = e$ and get rid of the clipping operator. Then, the first term becomes

$$
\begin{aligned}
&\mathbb{E}_{y^+,y^-} \left[ \lambda \log \frac{\pi_\theta(y^+ \mid \boldsymbol{c}_i)}{\pi_{\text{flt}}(y^+ \mid \boldsymbol{c}_i)} - \lambda \log \frac{\pi_\theta(y^- \mid \boldsymbol{c}_i)}{\pi_{\text{flt}}(y^- \mid \boldsymbol{c}_i)} \right] \\
=&\mathbb{E}_{y^+,y^-} \left[ \log \left( \exp \left( \lambda \log \frac{\pi_\theta(y^+ \mid \boldsymbol{c}_i)}{\pi_{\text{flt}}(y^+ \mid \boldsymbol{c}_i)} - \lambda \log \frac{\pi_\theta(y^- \mid \boldsymbol{c}_i)}{\pi_{\text{flt}}(y^- \mid \boldsymbol{c}_i)} \right) /Z \right) \right] + \log Z \\
=&\mathbb{E}_{y^+,y^-} \left[ \log \Pr(y^+ \succ y^- \mid \boldsymbol{c}_i) \right] + \log Z,
\end{aligned}
$$

where $\log Z$ is constant in parameter $\theta$. Comparing this equation with Rafailov et al. (2024) draws a connection between OVERTONE and DPO. The second term of Eq (17), on the other hand, is an additional penalty to push $\pi_\theta$ stay close to $\pi_{\text{flt}}$ by using a forward KL, which has been explored in preference learning (Wang et al., 2024a).

In conclusion, OVERTONE can be seen as an unbiased estimator of a special DPO problem. This completes our proof. □

## B. Implementation Details

### B.1. Hyperparameters used in KE

We present the implementation details of our algorithms. All of our experiments are run on EasyEdit (Wang et al., 2024e). In general, we tuned hyperparameters for each KE method basis *using to the base version*, if the default setting from EasyEdit showed noticable inferior performance. See below for more details.

**FT-M** used the following hyperparameters:

- On ZsRE, Wiki$_{\text{recent}}$, Wiki$_{\text{counterfact}}$, and WikiBio: default training parameters from EasyEdit for both LLaMA 2 and LLaMA 3.
- On MQuAKE: Layers to tune: (20,21,22,23,24). Learning rate: 1e-3. Others unchanged.

**LoRA** used the following hyperparameters:

- On ZsRE, Wiki$_{\text{recent}}$, Wiki$_{\text{counterfact}}$, and WikiBio: default training parameters from EasyEdit for both LLaMA 2 and LLaMA 3.
- On MQuAKE: LoRA rank: 12. Iteration numbers: 50. Others unchanged.

**MELO** used the following hyperparameters:

- We set initial radius for each code in the code-book to 60 for LLaMA 2, and 30 for LLaMA 3. Due to the fact that the default choice 0.1 was too small to retrieve any edited parameters for rephrased queries or reasoning.
- Others unchanged.

**WISE** used the following hyperparameters:

- On OVERTONE, we shrunk activation thresholds by 0.6 in consideration of the milder overfitting from our method. We didn't tune this shrinkage factor so it can be suboptimal. All other parameters used default values from EasyEdit.

- We removed data augmentation for better measure HTO influence. This led to significantly faster editing speed (around 5 times speedup).

**ROME** and **MEMIT** used default choices from EasyEdit.

Finally, OVERTONE is tuned on a KE model base and applied to both LLMs. We didn't tune hyper-parameters extensively, so below $\epsilon$ and $n$ can be suboptimal.

- FT-M: $\epsilon = 0.01$, $n = 0.5$ for $n\sigma$-filtering, $\lambda = 0.1$ for mixing.

- LoRA: $\epsilon = 0.05$, $n = 0.5$ for $n\sigma$-filtering, $\lambda = 0.1$ for mixing.

- MELO: $\epsilon = 0.05$, $n = 1$ for $n\sigma$-filtering, $\lambda = 0.1$ for mixing.

- WISE: $\epsilon = 0.05$, $n = 1$ for $n\sigma$-filtering, $\lambda = 0.1$ for mixing.

### B.2. MQuAKE Experiment Details

MQuAKE benchmark follows a different evaluation pipeline for Single-Hop and Multi-Hop reasoning questions (Zhong et al., 2023; Wang et al., 2024f) that checks the existence of ground truth answer in LLM's generation. Our evaluation rubric followed Zhong et al. (2023). We noted that the reliability of evaluation results heavily relies on the use of a good prompt, our prompts are given below.

- **Single-Hop questions**: we used 1-shot prompting to guide the model provide answers directly, the complete prompt is

> You are a helpful AI assistant. Answer questions directly.
> Always format your response as:
> Final answer: [concise and direct final answer]
> Question: Who is the spouse of the head of state in United States of America?
> Answer: Jill Biden
> Question: *# Single-Hop question related to the new knowledge #*
> Answer:

- **Multi-Hop questions**: Again we used 1-shot prompting to guide the model provide answers based on chain-of-thought (Wei et al., 2022), the complete prompt is

> You are a helpful AI assistant. For each question:
> 1. Break it down into simpler subquestions
> 2. Answer each subquestion step by step.
> 3. Use your answers to provide a final answer after "Final answer: "
> Always format your response as:
> Subquestion: [your subquestion]
> Generated answer: [your answer]
> Final answer: [concise and direct final answer]
> Question: Who is the spouse of the head of state in United States of America?
> Subquestion: Who is the head of state in United States of America?
> Answer: The head of state in United States of America is Joe Biden.
> Subquestion: Who is the spouse of Joe Biden?
> Answer: The spouse of Joe Biden is Jill Biden.
> Final answer: Jill Biden
> Question: *# Multi-Hop question related to the new knowledge #*

In generation, we set temperature to 0.1. The maximum length was 30 for Single-Hop questions, and 200 for Multi-Hop questions. **Chat templates** are applied.

## C. More Experiment Results

We present the complete Continual Editing results here. Note that sequence $T = 1$ reduces to Single Edit results, but we present them again for completeness.

*Table 4.* Continual Editing performance (LLaMA 2). WISE requires additional irrelevant data for training, which is only available in ZsRE benchmark.

| | ZsRE | | | | | Wiki$_{recent}$ | | | | Wiki$_{counterfact}$ | | | | WikiBio | | |
|---|---|---|---|---|---|---|---|---|---|---|---|---|---|---|---|---|
| | Rel. | Gen. | Por. | Loc. | Avg. | Rel. | Por. | Loc. | Avg. | Rel. | Por. | Loc. | Avg. | Rel. | Loc. | Avg. |
| | | | | | | | | $T = 1$ | | | | | | | | |
| | Rel. | Gen. | Por. | Loc. | Avg. | Rel. | Por. | Loc. | Avg. | Rel. | Por. | Loc. | Avg. | Rel. | Loc. | Avg. |
| ROME | 96.61 | 83.91 | 55.7 | 96.96 | 83.3 | 99.02 | 54.21 | 55.91 | 69.71 | 97.2 | 56.85 | 50.4 | 68.15 | 96.41 | 59.14 | 77.78 |
| MEMIT | 94.22 | 88.2 | 57.91 | 98.28 | 84.65 | 97.71 | 52.93 | 55.05 | 68.56 | 96.38 | 59.34 | 45.7 | 67.14 | 93.78 | 56.74 | 75.26 |
| FT-M | 99.75 | 99.33 | 54.32 | 93.01 | 86.60 | 100.0 | 62.93 | 45.92 | 69.62 | 100.0 | 74.7 | 54.86 | 76.52 | 100.0 | 90.04 | 95.02 |
| + Ours | 99.75 | 96.8 | 57.08 | 96.54 | 87.54 | 100.0 | 63.91 | 60.4 | 74.77 | 100.0 | 73.62 | 75.34 | 82.99 | 100.0 | 93.46 | 96.73 |
| LoRA | 100.0 | 100.0 | 23.34 | 30.44 | 63.45 | 100.0 | 55.41 | 28.29 | 61.23 | 100.0 | 71.92 | 9.99 | 60.64 | 100.0 | 48.84 | 74.42 |
| + Ours | 100.0 | 94.31 | 61.16 | 87.2 | 85.67 | 100.0 | 63.67 | 58.72 | 74.13 | 100.0 | 73.96 | 57.85 | 77.27 | 97.68 | 68.45 | 83.06 |
| MELO | 100.0 | 96.77 | 27.11 | 92.35 | 79.06 | 99.13 | 54.04 | 40.96 | 64.71 | 99.0 | 71.78 | 55.83 | 75.54 | 99.97 | 80.77 | 90.37 |
| + Ours | 100.0 | 93.31 | 50.36 | 97.2 | 85.22 | 100.0 | 60.25 | 66.48 | 75.58 | 99.91 | 71.81 | 78.09 | 83.27 | 99.68 | 82.58 | 91.13 |
| WISE | 92.42 | 70.86 | 54.57 | 100.0 | 79.46 | - | - | - | - | - | - | - | - | - | - | - |
| + Ours | 97.55 | 76.09 | 54.17 | 100.0 | 81.95 | - | - | - | - | - | - | - | - | - | - | - |
| | | | | | | | | $T = 10$ | | | | | | | | |
| | Rel. | Gen. | Por. | Loc. | Avg. | Rel. | Por. | Loc. | Avg. | Rel. | Por. | Loc. | Avg. | Rel. | Loc. | Avg. |
| ROME | 74.94 | 69.67 | 51.12 | 71.72 | 66.86 | 98.14 | 55.16 | 54.73 | 69.34 | 86.17 | 47.36 | 38.99 | 57.51 | 40.55 | 25.98 | 33.27 |
| MEMIT | 68.39 | 66.26 | 46.66 | 84.22 | 66.38 | 96.51 | 54.2 | 52.56 | 67.76 | 89.64 | 54.71 | 38.2 | 60.85 | 52.2 | 38.54 | 45.37 |
| FT-M | 89.14 | 87.43 | 47.13 | 84.26 | 76.99 | 97.4 | 56.47 | 41.4 | 65.09 | 96.41 | 70.32 | 42.44 | 69.72 | 92.96 | 77.69 | 85.32 |
| + Ours | 92.8 | 88.21 | 55.74 | 91.06 | 81.95 | 96.42 | 61.65 | 53.13 | 70.40 | 98.72 | 72.47 | 65.46 | 78.88 | 95.26 | 84.43 | 89.84 |
| LoRA | 29.25 | 30.41 | 19.83 | 24.81 | 26.07 | 35.17 | 23.8 | 24.98 | 27.98 | 22.64 | 13.87 | 10.24 | 15.58 | 70.45 | 46.82 | 58.64 |
| + Ours | 85.4 | 81.5 | 61.03 | 74.41 | 75.59 | 94.55 | 59.16 | 49.09 | 67.60 | 71.61 | 51.91 | 32.65 | 52.06 | 74.74 | 48.35 | 61.55 |
| MELO | 94.13 | 83.06 | 50.48 | 96.5 | 81.04 | 91.73 | 53.02 | 81.09 | 75.28 | 92.52 | 64.55 | 99.98 | 85.68 | 95.44 | 97.94 | 96.69 |
| + Ours | 94.38 | 81.89 | 54.92 | 98.41 | 82.40 | 91.69 | 54.95 | 93.22 | 79.95 | 93.49 | 63.36 | 99.98 | 85.61 | 95.24 | 97.77 | 96.50 |
| WISE | 84.5 | 73.81 | 53.19 | 100.0 | 77.88 | - | - | - | - | - | - | - | - | - | - | - |
| + Ours | 86.68 | 77.24 | 54.0 | 100.0 | 79.48 | - | - | - | - | - | - | - | - | - | - | - |
| | | | | | | | | $T = 100$ | | | | | | | | |
| | Rel. | Gen. | Por. | Loc. | Avg. | Rel. | Por. | Loc. | Avg. | Rel. | Por. | Loc. | Avg. | Rel. | Loc. | Avg. |
| ROME | 25.37 | 22.68 | 4.73 | 5.1 | 14.47 | 24.99 | 13.12 | 8.55 | 15.56 | 0.0 | 0.0 | 0.0 | 0.0 | 2.63 | 15.74 | 9.18 |
| MEMIT | 2.58 | 2.88 | 0.24 | 2.5 | 2.05 | 70.22 | 41.12 | 38.43 | 49.92 | 0.82 | 0.97 | 0.26 | 0.69 | 0.0 | 15.74 | 7.87 |
| FT-M | 88.36 | 84.51 | 41.76 | 54.11 | 67.19 | 97.51 | 53.73 | 33.88 | 61.71 | 95.69 | 66.23 | 26.69 | 62.87 | 93.56 | 67.51 | 80.53 |
| + Ours | 89.38 | 82.13 | 52.69 | 72.39 | 74.15 | 96.32 | 58.28 | 47.04 | 67.21 | 95.93 | 68.16 | 44.28 | 69.46 | 95.35 | 74.91 | 85.13 |
| LoRA | 0.67 | 0.78 | 1.00 | 0.03 | 0.62 | 0.5 | 0.5 | 0.12 | 0.37 | 0.67 | 0.0 | 0.0 | 0.22 | 47.02 | 27.06 | 37.04 |
| + Ours | 62.23 | 58.06 | 56.62 | 59.57 | 59.12 | 70.49 | 47.05 | 49.87 | 55.80 | 32.17 | 28.99 | 29.19 | 30.12 | 52.96 | 25.73 | 39.34 |
| MELO | 38.13 | 36.12 | 53.88 | 98.08 | 56.55 | 26.33 | 24.98 | 53.73 | 35.01 | 24.87 | 24.21 | 78.71 | 42.60 | 48.88 | 97.61 | 48.88 |
| + Ours | 39.13 | 37.28 | 54.75 | 98.58 | 57.44 | 47.95 | 39.65 | 86.77 | 58.12 | 24.92 | 25.39 | 97.12 | 49.14 | 52.17 | 97.44 | 74.81 |
| WISE | 84.59 | 71.59 | 54.45 | 100.0 | 77.66 | - | - | - | - | - | - | - | - | - | - | - |
| + Ours | 92.42 | 84.22 | 56.71 | 100.0 | 83.34 | - | - | - | - | - | - | - | - | - | - | - |

# D. More Discussions

We discuss some more conceptual characteristics and potential problems that future works may work on here.

**OVERTONE and ROME/MEMIT.** ROME (Meng et al., 2022a) and MEMIT (Meng et al., 2022b) are representative solutions of KE through the locate-and-editing paradigm that are built upon the causal tracing and explicitly constructed updated rules. This leads to special KE losses which contains two unique designs other than those being used in the four backbone methods we studied. First, the impact of auto-regressive loss, which OVERTONE alters, on ROME is weaker, in the sense that the MSE loss will determine the final parameter update. Second, ROME relies on random prefix augmentation, which affects overfitting as well. Given these facts, we plan to work on a more principled way to extend OVERTONE, a augmentation-free end-to-end training paradigm, in light of its principle. That is, we seek a better way to smooth (relax) different token fitting adaptively with the model's own knowledge, following the principle of OVERTONE. Therefore, it would be interesting to bring the idea of OVERTONE to training ROME and MEMIT to boost their generalizability.

*Table 5.* Continual Editing performance (LLaMA 3). WISE requires additional irrelevant data for training, which is only available in ZsRE benchmark.

| | ZsRE | | | | | Wiki$_{recent}$ | | | | Wiki$_{counterfact}$ | | | | WikiBio | | |
|---|---|---|---|---|---|---|---|---|---|---|---|---|---|---|---|---|
| | **Rel.** | **Gen.** | **Por.** | **Loc.** | **Avg.** | **Rel.** | **Por.** | **Loc.** | **Avg.** | **Rel.** | **Por.** | **Loc.** | **Avg.** | **Rel.** | **Loc.** | **Avg.** |
| | | | | | | | | $T = 1$ | | | | | | | | |
| **ROME** | 99.17 | 97.91 | 58.12 | 95.9 | 87.78 | 98.84 | 54.76 | 49.74 | 67.78 | 99.94 | 58.0 | 42.94 | 66.96 | 92.43 | 72.63 | 82.53 |
| **MEMIT** | 96.67 | 92.46 | 58.78 | 98.23 | 86.53 | 98.51 | 53.65 | 48.45 | 66.87 | 99.44 | 57.81 | 42.73 | 66.66 | 96.26 | 71.23 | 83.75 |
| **FT-M** | 100.0 | 99.75 | 40.43 | 79.43 | 79.90 | 100.0 | 57.13 | 30.01 | 62.38 | 100.0 | 72.62 | 31.47 | 68.03 | 100.0 | 92.96 | 96.48 |
| **+ Ours** | 100.0 | 99.75 | 48.63 | 94.78 | 85.79 | 100.0 | 60.88 | 44.67 | 68.52 | 100.0 | 73.5 | 58.29 | 77.26 | 99.99 | 94.87 | 97.43 |
| **LoRA** | 100.0 | 100.0 | 26.55 | 38.85 | 66.35 | 100.0 | 52.99 | 26.46 | 59.82 | 100.0 | 71.1 | 9.02 | 60.04 | 100.0 | 59.77 | 79.88 |
| **+ Ours** | 100.0 | 98.5 | 51.57 | 93.13 | 85.80 | 100.0 | 61.46 | 56.1 | 72.52 | 100.0 | 72.8 | 57.54 | 76.78 | 98.16 | 77.24 | 87.7 |
| **MELO** | 100.0 | 96.84 | 39.63 | 98.8 | 83.82 | 100.0 | 59.07 | 65.78 | 74.95 | 100.0 | 71.55 | 87.77 | 86.44 | 100.0 | 98.56 | 99.28 |
| **+ Ours** | 100.0 | 95.77 | 43.08 | 98.8 | 84.41 | 100.0 | 58.72 | 69.1 | 75.94 | 100.0 | 70.26 | 89.81 | 86.69 | 99.98 | 98.56 | 99.27 |
| **WISE** | 71.67 | 51.29 | 49.27 | 100.0 | 68.06 | - | - | - | - | - | - | - | - | - | - | - |
| **+ Ours** | 82.67 | 62.34 | 47.54 | 100.0 | 73.14 | - | - | - | - | - | - | - | - | - | - | - |
| | **Rel.** | **Gen.** | **Por.** | **Loc.** | **Avg.** | **Rel.** | **Por.** | **Loc.** | **Avg.** | **Rel.** | **Por.** | **Loc.** | **Avg.** | **Rel.** | **Loc.** | **Avg.** |
| | | | | | | | | $T = 10$ | | | | | | | | |
| **ROME** | 43.91 | 40.14 | 25.11 | 31.7 | 35.22 | 91.17 | 51.25 | 43.67 | 62.03 | 86.52 | 45.37 | 32.9 | 54.93 | 4.01 | 7.58 | 5.79 |
| **MEMIT** | 59.74 | 58.36 | 37.34 | 71.06 | 56.62 | 98.38 | 54.42 | 47.08 | 66.63 | 98.61 | 58.48 | 36.28 | 64.46 | 5.4 | 1.61 | 3.5 |
| **FT-M** | 79.54 | 78.44 | 25.03 | 43.97 | 56.75 | 87.22 | 48.12 | 25.8 | 53.71 | 90.13 | 62.37 | 13.83 | 55.44 | 95.59 | 87.45 | 91.52 |
| **+ Ours** | 84.74 | 81.41 | 44.2 | 75.67 | 71.50 | 92.77 | 52.65 | 38.99 | 61.47 | 93.04 | 66.5 | 39.99 | 66.51 | 96.81 | 91.17 | 93.99 |
| **LoRA** | 18.54 | 17.55 | 6.63 | 6.56 | 12.32 | 21.7 | 13.66 | 11.97 | 15.78 | 12.59 | 5.92 | 0.69 | 6.40 | 51.09 | 44.45 | 47.77 |
| **+ Ours** | 73.28 | 72.39 | 53.13 | 69.36 | 67.04 | 93.68 | 56.97 | 49.34 | 66.66 | 71.99 | 49.52 | 32.24 | 51.25 | 64.26 | 55.11 | 59.69 |
| **MELO** | 94.08 | 80.47 | 47.97 | 98.8 | 80.33 | 92.56 | 54.51 | 86.58 | 77.88 | 92.97 | 63.74 | 98.3 | 85.00 | 94.77 | 98.56 | 96.67 |
| **+ Ours** | 94.08 | 80.94 | 49.77 | 98.8 | 80.90 | 91.56 | 54.24 | 89.16 | 78.32 | 92.97 | 62.69 | 98.32 | 84.66 | 94.91 | 98.56 | 96.74 |
| **WISE** | 51.14 | 43.36 | 51.0 | 100.0 | 61.38 | - | - | - | - | - | - | - | - | - | - | - |
| **+ Ours** | 58.21 | 53.22 | 49.21 | 100.0 | 65.16 | - | - | - | - | - | - | - | - | - | - | - |
| | **Rel.** | **Gen.** | **Por.** | **Loc.** | **Avg.** | **Rel.** | **Por.** | **Loc.** | **Avg.** | **Rel.** | **Por.** | **Loc.** | **Avg.** | **Rel.** | **Loc.** | **Avg.** |
| | | | | | | | | $T = 100$ | | | | | | | | |
| **ROME** | 7.18 | 6.02 | 1.04 | 2.24 | 4.12 | 8.89 | 1.36 | 0.31 | 3.52 | 3.92 | 0.99 | 0.0 | 1.64 | 0.88 | 7.47 | 4.18 |
| **MEMIT** | 0.0 | 0.0 | 0.0 | 0.0 | 0.0 | 0.57 | 0.92 | 0.4 | 0.63 | 0.81 | 0.86 | 0.0 | 0.56 | 0.01 | 23.44 | 11.73 |
| **FT-M** | 78.79 | 78.29 | 13.7 | 15.42 | 46.55 | 94.27 | 44.09 | 22.99 | 53.78 | 87.47 | 55.62 | 2.78 | 48.62 | 93.65 | 85.83 | 89.74 |
| **+ Ours** | 81.2 | 77.87 | 32.65 | 44.66 | 59.09 | 96.19 | 53.73 | 32.42 | 60.78 | 92.97 | 62.02 | 20.71 | 58.57 | 94.23 | 85.83 | 94.23 |
| **LoRA** | 1.75 | 1.81 | 1.29 | 2.13 | 1.74 | 1.33 | 1.58 | 0.93 | 1.28 | 1.00 | 0.00 | 0.00 | 0.33 | 15.88 | 17.61 | 16.74 |
| **+ Ours** | 51.38 | 50.3 | 49.72 | 35.83 | 46.81 | 64.82 | 42.92 | 44.27 | 50.67 | 25.31 | 20.18 | 17.49 | 20.99 | 19.03 | 10.9 | 14.96 |
| **MELO** | 29.79 | 28.83 | 50.01 | 98.8 | 51.86 | 36.71 | 29.02 | 83.23 | 49.65 | 22.2 | 22.9 | 97.85 | 22.55 | 52.19 | 98.56 | 75.37 |
| **+ Ours** | 29.79 | 28.73 | 50.01 | 98.8 | 51.83 | 40.42 | 34.85 | 92.67 | 55.98 | 22.45 | 22.9 | 97.85 | 47.73 | 52.15 | 98.56 | 75.36 |
| **WISE** | 84.87 | 74.87 | 39.24 | 100.0 | 74.75 | - | - | - | - | - | - | - | - | - | - | - |
| **+ Ours** | 86.83 | 77.54 | 34.99 | 100.0 | 74.84 | - | - | - | - | - | - | - | - | - | - | - |

**Potential Bias in OVERTONE Design.** OVERTONE is designed to the model's own prediction to extract pretrained knowledge that should be maintained. To avoid misleading knowledge conflict and general noise, OVERTONE incorporates two mechanisms. First, the unreliable (noisy) part is filtered out. Second, mixing with the model's prediction is conducted only if the mixed distribution correctly assigns the ground truth label (i.e., training token) the highest probability. However, *provably* solving the potential knowledge conflict and identifying the optimal target distribution for KE are still two open questions, and we advocate for future studies to work on these two directions towards better KE.

