# OpenReview forum: "Mitigating Heterogeneous Token Overfitting in LLM Knowledge Editing"
_ICML.cc/2025/Conference — ICML 2025 poster_

### Official Review · Reviewer_drcx · 2025-02-21

**Overall Recommendation:** 4

**Summary:**

This paper addresses the problem of heterogeneous token overfitting (HTO) in knowledge editing (KE) for large language models (LLMs). The authors identify that existing KE methods, which indiscriminately optimize cross-entropy loss across all tokens, lead to varying overfitting rates for different tokens, degrading reasoning capabilities. They propose OVERTONE, a token-level smoothing method that adaptively refines target distributions by blending ground-truth tokens with filtered model predictions. Experiments across four KE methods, two LLMs, and diverse benchmarks demonstrate OVERTONE’s effectiveness in improving portability and locality while maintaining reliability.

## update after rebuttal
I keep my score as the final score.

**Claims And Evidence:**

Yes

**Essential References Not Discussed:**

There are some answer-level overfitting analyses that I think should be mentioned. Token level overfitting is good but answer-level is also important.

- Neighboring Perturbations of Knowledge Editing on Large Language Models, ICML 2024

**Experimental Designs Or Analyses:**

Yes
- I do not understand the relation between the portability loss and the underfitting degree? Why the underfitting degree prove the model is overfit? It is not clear hear.

**Methods And Evaluation Criteria:**

Yes

**Other Comments Or Suggestions:**

NA

**Other Strengths And Weaknesses:**

Strengths:
- The analysis is good and tackles the issues well.
- The experiments are adequate and convincing.
- The identification of HTO as a key bottleneck in KE is a significant contribution. The analysis of token-level loss dynamics provides a fresh perspective on overfitting in LLM editing.

Weakness:

- While the connection to DPO is intriguing, the paper does not empirically compare OVERTONE with DPO-based editing methods, leaving its practical advantages under-explored.
- Some case studies could make the contribution clearer.

**Questions For Authors:**

Some questions:

- I remember the MQuAKE contains 2-4 hops questions; why does you just do the 2-hops?

**Relation To Broader Scientific Literature:**

This work advances knowledge editing (KE) by addressing token-level overfitting—a gap in prior KE methods (e.g., MEND, LoRA) that optimize generically across tokens. By integrating token-aware regularization and influence-function theory, it bridges fine-grained training dynamics with LLM robustness, offering a universal enhancement for KE frameworks.

**Theoretical Claims:**

Yes

---

> ### Author Rebuttal · Authors · 2025-03-31
>
> We highly appreciate your effort and time spent reviewing our paper and thank you for your expertise and constructive comments. In the following, we address your comments and questions one by one.
>
> >**The relation between the portability loss and the underfitting degree (UD).**
>
> Yes, portability loss does not directly imply UD. In Sec 2.2, Fig 1 shows that high portability loss is attributed to overfitting. Toward a deeper understanding of such overfitting, in Sec 2.3, (negative) UD (NUD) is defined to uncover the token-level HTO, which is a key contribution of this work as admired by the reviewer.
>
> Specifically, NUD represents a token is overfitted as its training loss is too small. Specifically, NUD computes the difference between the token's training loss (of edited model), and the greedy decoded token's pretrained loss (of unedited model). The choice of greedy decoding is on purpose, as it reflects the unedited model's most confident knowledge proper and valid in the past. By comparing the two, NUD indicates that the edited model is overly confident, and is "overfitted" thereof. We will make this clearer.
>
> >**Related work on Answer-level overfitting.**
>
> We thank the reviewer for bringing this interesting work to our attention. After reading it, we agree that the neighboring knowledge perturbation due to the answer-level overfitting is insightful, and it would be interesting to explore bridging the two types of overfitting and building more principled solutions. We will highlight this referred paper, the connection, and this future direction in the revision.
>
> >**Empirical Comparison with DPO.**
>
> Following the reviewer's suggestion,
> we train LoRA with DPO from EasyEdit.
> We use the pre-edited model's old knowledge as the negative data.
> Due to time constraint, we only conduct Single Editing on ZsRE.
> We note that DPO performs worse, which we presume due to the practical challenge analyzed in Sec 3.2.
> }
>
> |        |       | Rel. | Gen.  | Por.  | Loc.  | Avg   |
> |--------|-------|------|-------|-------|-------|-------|
> | Llama2 | Ours  |  100 | 94.31 | 61.16 |  87.2 | 85.67 |
> |        | DPO   |  100 | 94.74 | 33.64 | 41.66 | 67.51 |
> | Llama3 | Ours  |  100 |  98.5 | 51.57 | 93.13 |  85.8 |
> |        | DPO   |  100 | 97.77 | 19.61 | 10.58 | 56.99 |
>
> >**Some case studies could make the contribution clearer.**
>
> Thank you very much for the suggestion.
> Due to time constraint, we will dive into more in-depth visualization on OVERTONE effect, and how OVERTONE helps multi-hop reasoning on MQuAKE in the revised paper.
>
> >**MQuAKE contains 2-4 hops questions; why does you just do the 2-hops?**
>
> We apologize for the misleading statement. In experiments we followed MQuAKE official Repo, DeepEdit, and EasyEdit to load the multi-hop questions. We didn't manually filter out 3 and 4 hops questions. After rechecking the source documents carefully, we noted that "2-Hop" is inaccurate and should be "Multi-Hop". We will correct these descriptions in the revised paper.

---

### Official Review · Reviewer_h3rV · 2025-03-07

**Overall Recommendation:** 3

**Summary:**

This paper investigates the Heterogeneous Token Overfitting problem in knowledge editing. The authors first analyze the root cause of this issue, attributing it to the training paradigm that indiscriminately optimizes the probabilities of all tokens. To address this, they propose OVERTONE, which refines the traditional loss function. The theoretical advantages of OVERTONE are demonstrated, and experiments show that it outperforms several baselines across diverse experimental settings.

**Claims And Evidence:**

The claims are supported by evidence.

**Essential References Not Discussed:**

No necessary related work has been omitted.

**Experimental Designs Or Analyses:**

1. I think OVERTONE can be applied to ROME (or MEMIT) to improve the loss function (Equation (4) in the original paper https://arxiv.org/pdf/2202.05262), but the experiment did not show the corresponding results.

2. This paper lacks a comparison with LTI ([1]), another method designed to alleviate overfitting.

3. The impact of varying the parameter filtering threshold n on the performance of the proposed method is not explored in the experiments.

References:
[1] https://openreview.net/forum?id=t8qcGXaepr

**Methods And Evaluation Criteria:**

While OVERTONE effectively adjusts the target distribution by filtering out noise tokens, it may inadvertently introduce bias in scenarios such as knowledge conflicts, where the model's own predicted distribution could be unreliable.

**Other Comments Or Suggestions:**

Please see above comments.

**Other Strengths And Weaknesses:**

Please see above comments.

**Questions For Authors:**

Recent research has begun exploring knowledge editing in the form of free text. Have the authors considered the problem of overfitting in this context?

**Relation To Broader Scientific Literature:**

Prior work has identified the problem of overfitting in knowledge editing. The main contribution of this paper is to analyze this problem from the token-level, providing new insights into its underlying causes. The authors propose OVERTONE, a new method to mitigate overfitting. This work advances the field by offering more granular understanding of the problem.

**Theoretical Claims:**

I have checked the correctness of any proofs for theoretical claims.

---

> ### Author Rebuttal · Authors · 2025-03-31
>
> We highly appreciate your effort and time spent reviewing our paper and thank you for your expertise and constructive comments. In the following, we address your comments and questions one by one.
>
> >**OVERTONE can be applied to ROME (or MEMIT) to improve the loss function (Eq 4 in the original paper).**
>
> Thank you for the insightful idea of extending OVERTONE to ROME. We noted two unique designs in ROME (and MEMIT) makes it differ from four methods we studied. First, the impact of auto-regressive loss, which OVERTONE alters, on ROME is weaker, in the sense that the MSE loss will determine the final parameter update. Second, ROME relies on random prefix augmentation, which affects overfitting as well. Given these facts, we plan to work on a more principled way to extend OVERTONE, a augmentation-free end-to-end training paradigm, in light of its principle. That is, we seek a better way to *smooth (relax) different token fitting adaptively* with the model's own knowledge, following the principle of OVERTONE. We will highlight this challenge, together with our future plan in the revision.
>
> >**Comparison with LTI, another method designed to alleviate overfitting.**
>
> Both LTI and OVERTONE works on mitigating overfitting in knowledge editing. The conceptual similarity, from a high level, lies in adding pre-trained knowledge to the editing.
> But LTI explores a distinct direction, with its difference to ours lying in X folds.
> First, LTI explores in-context learning (ICL) to incorporate pre-trained knowledge into the editing data,
> while ours designs an adaptive token-level distribution mixing, in light of the token-level HTO dynamic.
> Second, LTI, which is primarily developed for ROME-based solution, acts on both latent representation and output prediction loss. Ours, on the other hand, is agnostic to the editing method and alters the output prediction loss only.
> Finally, LTI, same as ROME, relies on a data augmentation, while ours does not include such mechanism.
> Following the reviewer's suggestion,
> we will highlight these differences in the revised paper, and will explore bridging the two directions in our future work.
>
> >**Bias in knowledge conflicts and the model's own predicted distribution could be unreliable.**
>
> We agree that potential knowledge conflict and general noise can be misleading.
> To reduce this risk, OVERTONE incorporates two mechanisms.
> First, the unreliable (noisy) part is filtered out.
> Second, mixing with the model's prediction is conducted only if the mixed distribution correctly assigns the ground truth label (i.e., training token) the highest probability (Eq 3).
> Finally, *provably solving the potential knowledge conflict* for knowledge editing is still an open question, and we will highlight this in the revision.
>
> >**Impact of filtering threshold $n$.**
>
> Mathematically, "without filtering" is equivalent to setting $n \rightarrow \infty$.
> As shown in Tab 3, this leads to a worse performance.
> To further study how sensitive $n$ is,
> we follow the reviewer's suggestion and try a larger $n=1$ on LoRA, which is the default value from Top $n\sigma$ paper.
> This gives us average performance 85.49 (Rel: 100, Gen: 94.85, Por: 61.44, Loc: 87.01) on editing ZsRE, which is slightly lower than 85.67 from $n=0.5$.
> We believe this insensitivity is reasonable, considering that correctness-checking mechanism will discard the mixing if it is misleading.
> We will add this discussion in the revised paper.
>
> >**Knowledge editing in free text form.**
>
> We thank the reviewer for coming up this interesting direction.
> After checking related papers, we agree that the free-form text can express more diverse knowledge, on which the editing can be important but also more challenging. Considering the common practice that knowledge to edit at a time is few, we expect a similar overfitting due to the small training size nature, and HTO because some pretrained knowledge can still be useful, making different parts (tokens) have varied difficulties to learn. Therefore, we believe that our method can shed light on this interesting problem.
> However, we also believe that the free-form will add additional challenge to understand and quantify the overfitting.
> Therefore, we will explore this direction in our future work, and add this discussion, together with related papers, in the revised paper.

---

### Official Review · Reviewer_Dqfx · 2025-03-15

**Overall Recommendation:** 3

**Summary:**

This paper investigates the challenge of heterogeneous token overfitting in knowledge editing of large language models, where different tokens in the target knowledge generalize at varying rates during selective parameter updates. To address this, the authors propose OVERTONE—a token-level smoothing approach that adaptively refines the training target for each token to mitigate overfitting while preserving unrelated pre-trained knowledge. The paper presents both a detailed theoretical analysis, which connects OVERTONE to concepts such as DPO, and extensive empirical evaluations on multiple benchmarks using models like LLaMA 2 and LLaMA 3. The results demonstrate that OVERTONE significantly enhances editing performance by improving the model’s reasoning (portability), generality, and locality with negligible computational overhead, offering a flexible plug-and-play solution that can complement existing KE methods.

## update after rebuttal
The additional experiments have indeed resolved most of my doubts and provided more comprehensive support for the arguments in this paper. I am inclined to accept this paper, but since I initially gave it a score of 3, which means leaning towards accept, I will maintain the score of 3.

**Claims And Evidence:**

The paper's core claims about the effectiveness of the OVERTONE method are generally well-supported by extensive experimental evidence across multiple editing methods, models, and datasets. However, some claims lack sufficient supporting evidence:

- Model-agnostic wide applicability: While the effectiveness has been demonstrated across 4 editing methods, testing was limited to only two LLaMA series models. Experiments on additional architectures such as Qwen or Mistral would strengthen the validation of the method's broad applicability.

- Connection to DPO: The derivation linking OVERTONE to DPO is mathematically detailed; however, the practical implications of this connection are not supported by direct experimental comparisons. More empirical evidence demonstrating that the benefits of DPO carry over to the OVERTONE framework would help solidify this claim.

**Essential References Not Discussed:**

I think the paper adequately covers the essential references needed to understand the context for its key contributions. But I am not very familiar with this domain.

**Experimental Designs Or Analyses:**

The experimental designs and analyses were carefully structured and exhibit strong soundness and validity. However, there are some points which can be improved:

- Although the experimental results demonstrate significant improvements across various metrics, the study would benefit from deeper statistical validation. Specifically, including statistical significance tests along with reporting the mean and variance from multiple experimental runs (e.g., using different random seeds) would enhance the robustness and persuasiveness of the conclusions.

- Additionally, the experiments predominantly rely on LLaMA 2 and LLaMA 3, which share substantial architectural similarities. This architectural limitation may affect the generalizability of the method to other model designs, such as Qwen and Mistral, or multimodal models. Expanding the evaluation to include a more diverse set of architectures would help in thoroughly validating the broader applicability of the proposed method.

**Methods And Evaluation Criteria:**

The proposed methods and evaluation criteria are appropriate for the problem at hand; I believe that aside from not considering a broader range of model architectures, both the approach and evaluations are reasonable.

**Other Comments Or Suggestions:**

- Consider including statistical significance tests along with mean and variance from multiple runs (e.g., using different random seeds) to further validate the improvements and robustness of OVERTONE.
Broader Model Evaluation:
- Expanding the experimental evaluation to include architectures beyond LLaMA 2 and LLaMA 3 (such as Qwen or Mistral) would strengthen the claim regarding the method's model-agnostic benefits.

**Other Strengths And Weaknesses:**

The paper introduces a novel token-level adaptive smoothing approach that effectively mitigates heterogeneous token overfitting in knowledge editing for large language models. The method is supported by thorough theoretical derivations and extensive empirical evaluations on LLaMA 2 and LLaMA 3, showing significant improvements in reasoning capacity, generality, and locality.

However, the experimental validation is limited to a pair of closely related architectures, suggesting that further statistical validation and broader model evaluations are needed to confirm the method’s wide applicability and real-world effectiveness.

**Questions For Authors:**

- Your experiments primarily focus on LLaMA 2 and LLaMA 3 models. Have you evaluated or can you comment on how your approach scales to other architectures (e.g., Qwen or Mistral)?
- Could you provide details on how the hyperparameters were selected for the experiments and discuss any observed sensitivity to these choices?
- The paper draws a connection between your approach and DPO. Could you provide more insights or quantitative analyses on how this theoretical connection affects the overall performance of the OVERTONE method in practice?
- While the experimental results show promising improvements, could you elaborate on the variability of these results? Specifically, have you conducted multiple runs with different random seeds and performed statistical significance tests to verify the robustness of your improvements?

**Relation To Broader Scientific Literature:**

This paper tackles the issue of heterogeneous token overfitting in knowledge editing for large language models by building on prior work in areas like selective fine-tuning and parameter-efficient methods (e.g., LoRA, ROME, and MEMIT). It extends traditional strategies such as label smoothing and early stopping by introducing a token-level adaptive smoothing approach that preserves the model’s pre-trained knowledge while integrating new information. The authors support their method with a theoretical analysis based on influence functions and draw connections to constrained optimization techniques like direct preference optimization (DPO). Overall, the approach not only improves the reliability, generalizability, and locality of edited models but also offers a versatile, model-agnostic framework that advances both the practical and theoretical understanding of LLM knowledge editing.

**Theoretical Claims:**

I reviewed the proofs for all the proposition. In general, the derivations appear logically align with standard techniques such as Taylor expansion, KL-divergence properties, and influence function analysis. However, several proofs rely on nontrivial assumptions—for example, assumptions about gradient isotropy, the convergence of pretrained models, and specific cosine similarity bounds—which may be strong in practice. These assumptions, along with simplifications in the first-order approximations and the filtering mechanism provide valuable theoretical intuition, further empirical validation.

---

> ### Author Rebuttal · Authors · 2025-03-31
>
> We highly appreciate your effort and time spent reviewing our paper and thank you for your expertise and constructive comments. In the following, we address your comments and questions one by one.
> >**More model architecture.**
>
> We follow recent works (e.g. EasyEdit survey) to study the representative LLaMA family. Following the reviewer's suggestion,
> we further study Qwen2.5-3B-Instruct. Due to time constraint, we only experiment it with ZsRE, using hyperparameters on LLaMA-2 (which can be suboptimal).
> As shown at: https://anonymous.4open.science/r/hto-overtone, OVERTONE again helps achieve better editing performance. For the reviewer's convenience, we paste FT-M Single Edit results here.
> |        | Rel. | Gen.  | Por.  | Loc.  | Avg   |
> |--------|------|-------|-------|-------|-------|
> | FT-M   |  100.0 |  99.3 | 50.98 | 73.13 | 80.85 |
> | +Ours  |  100.0 | 96.18 | 56.26 | 80.66 | 83.28 |
>
> >**Comparison with DPO.**
>
> Following the reviewer's suggestion, we train LoRA with DPO.We use the pre-edited model's old knowledge as the negative data. Due to time constraint, we only conduct Single Editing on ZsRE. We note that DPO performs worse, which we presume due to the practical challenge highlighted in Sec 3.2.
>
> |        |       | Rel. | Gen.  | Por.  | Loc.  | Avg   |
> |--------|-------|------|-------|-------|-------|-------|
> | Llama2 | Ours  |  100 | 94.31 | 61.16 |  87.2 | 85.67 |
> |        | DPO   |  100 | 94.74 | 33.64 | 41.66 | 67.51 |
> | Llama3 | Ours  |  100 |  98.5 | 51.57 | 93.13 |  85.8 |
> |        | DPO   |  100 | 97.77 | 19.61 | 10.58 | 56.99 |
>
> >**Statistical significance tests along with mean and variance from multiple runs.**
>
> Our experiment design follows the convention in Knowledge Editing and is based on the widely-used EasyEdit.
> All metrics are average from different samples, each uses a *different* initial value. The random seed is fixed to 42, so "standard" and "ours" use the identical initial value for the same sample.
> Following the reviewer's thought,
> we tried seed 2025, and ran "ours" as in Tab 3, resulting in a new average 85.44 (Rel 100, Gen 94.88, Por 60.41, Loc 86.47),
> which is very close to the reported one, confirming the effectiveness of our method.
> From a statistical test perspective, out of 72 comparisons ("standard" vs "ours"),
> ours achieved better performance in 69 cases, providing a "significant" improvement based on binomial test.
>
> Finally, we agree with the reviewer that conducting multiple runs on each sample (knowledge) will further enhance the reliability, which is valuable but has been largely overlooked by the community.
> We will highlight this in the limitation section of the revised paper, and will follow this principle in our future work.
>
> >**Hyperparameter selection and sensitivity.**
>
> We didn't conduct an extensive hyperparameter tuning. Our current selection can be found in App B, and are made as follows: $\epsilon$ is set to close to 0, and we tried 0.05 and 0.01; $n$ for filtering tried $1$ (default in "Top $n\sigma$" paper) and more aggressive $0.05$ for simpler LoRA and FT. Finally, mixing $\lambda$ was set to $0.1$ to encourage fast integration of $\pi_{flt}$ without tuning. We didn't notice too big difference when trying different $n$, which we found reasonable, as the *correctness-checking mechanism (Eq 4), will discard a smoothing if it is too misleading*. However, OVERTONE can be sensitive to $\lambda$, as greater $\lambda$ makes our method more similar to standard training.

---

### Official Review · Reviewer_zvke · 2025-03-16

**Overall Recommendation:** 3

**Summary:**

This paper proposes OVERTONE, a token-level smoothing method to address heterogeneous token overfitting (HTO) in knowledge editing (KE) for large language models (LLMs), enabling specific knowledge updates without compromising pre-trained capabilities. Experiments across multiple methods, LLMs, and scenarios show OVERTONE improves performance and versatility over previous KE approaches, with minimal computational overhead and an implicit DPO mechanism.

**Claims And Evidence:**

The claims made in the submission are supported by clear and convincing evidence.

**Essential References Not Discussed:**

I think there are no essential related works missing from the paper that are critical to understanding the context of its key contributions.

**Ethical Review Concerns:**

N.A.

**Experimental Designs Or Analyses:**

The soundness and validity of the experimental designs and analyses seem to be appropriate.

**Methods And Evaluation Criteria:**

The proposed methods and evaluation criteria make sense for the problem at hand.

**Other Comments Or Suggestions:**

N.A.

**Other Strengths And Weaknesses:**

**Strength:**

* The paper is well written with clear motivations.

* The paper conducts comprehensive experiments.

**Weakness:**

* Experimental results, particularly in Table 5 and Table 1, raise doubts due to lower portability compared to WISE and generalization declines in FT/LoRA. Please see questions.

**Questions For Authors:**

1. What is meant by "We define underfitting degree (UD) as the difference between the pre-edited and running log-likelihood; negative UD indicates an overfitting"? Why does a negative UD represent overfitting?

2. Is the proposed solution (OVERTONE) adaptable to other fine-tuning methods, tasks, and datasets beyond knowledge editing tasks? Additionally, does it still perform well on long-text scenarios?

3. Why does portability, an emphasized metric for overfitting in this paper, perform significantly worse than the WISE method in many scenarios (e.g., Table 5)? Additionally, why is there a notable decline in generalization for FT and LoRA?

4. Why were methods like MEND, MEMIT, and MELLO not evaluated in Table 2?

5. In Table 3, why does adding filtering tail regions lead to a decrease in locality? Does this imply that some useful general information is being filtered out?

6. Minor: There is a grammatical error in lines 314-315: “*We next check where the improvement was made. from the table, the first gain was from improved portability.*”

**Relation To Broader Scientific Literature:**

The key contributions of the paper build on prior work in knowledge editing (KE) and large language model (LLM) fine-tuning. The identification of heterogeneous token overfitting (HTO) as a critical issue in KE extends the understanding of overfitting in LLMs, which has been explored in works such as Zhang et al. (2024), who investigated overfitting in fine-tuning LLMs.

**Theoretical Claims:**

I did not check the theorectical claim thoroughly.

---

> ### Author Rebuttal · Authors · 2025-03-31
>
> We highly appreciate your effort and time spent reviewing our paper and thank you for your expertise and constructive comments. In the following, we address your comments and questions one by one.
> >**Why does negative UD (NUD) represent overfitting?**
>
> NUD represents a *token* is overfitted as its training loss is *too small*. Specifically, NUD computes the difference between the token's training loss (of edited model), and the *greedy decoded* token's *pretrained* loss (of unedited model). The choice of greedy decoding is on purpose, as it reflects the unedited model's *most confident knowledge proper and valid in the past*. By comparing the two, NUD indicates that the edited model is overly confident, and is "*overfitted*" thereof. We will make this clearer.
> >**Is OVERTONE adaptable to other fine-tuning methods and tasks? On long-text scenarios?**
>
> OVERTONE is developed in light of HTO (i.e., editing knowledge is overfitted at different speeds). As per Sec 2.3, one cause of HTO is when knowledge editing (KE) involves few training data (such as single one) and trains the model with the fixed data for many steps, inevitably overfits. Similar concerns may raise in other tasks that seek *selective* updates of LLMs such as machine unlearning, wherefore OVERTONE could be applied. Moreover, when the training text is long, as the number of tokens to learn grows, we expect HTO to exacerbate, and OVERTONE to be helpful.
> >**Why does portability perform significantly worse on WISE (Table 5)?**
>
> Table 5 reports editing performance on LLaMA 3 with hyperparameter adopted from LLaMA 2, which can be suboptimal. As an evidence, the vanilla WISE also performs worse than on LLaMA 2 (WISE has an activation mechanism to determine if edited parameters should be used, which needs additional tuning as well). Nonetheless, OVERTONE was able to help achieve a better editing-generality-portability trade-off, leading to a higher *average* performance.
> >**Why a notable decline in generalization (gen) for FT and LoRA?**
>
> We believe the generality (Gen) decrease is also caused by suboptimal hyperparameter: As shown Fig 1, Gen doesn't encounter degradation due to HTO: as Gen loss change is nearly identical to that of training loss. Therefore, when OVERTONE mitigates HTO, it slightly decreases Gen while achieving better portability and locality. Still, the *average* performance is improved, and we believe our method can benefit from a more extensive hyperparameter tuning.
> >**Why were MEND, MEMIT, and MELLO not evaluated in Table 2?**
>
> Table 2 showed that OVERTONE can help improve reasoning in more challenging scenarios. Therefore, we focused on FT and LoRA, two simple methods *suffered more degradation* from HTO, to demonstrate the effectiveness of OVERTONE. We didn't include MEND, MEMIT, and MELLO because of their different mechanisms: MEND relies on a *large external dataset* to train its hypernet, MEMIT uses a objective other than maximizing the editing data likelihood, and MELLO is training-free.
> >**Filtering tail regions makes locality decrease.**
>
> This trend can be related to the definition of locality. Conceptually, a perfect locality only requires the edited model prediction match its pretrained output, regardless of its correctness and usefulness. Therefore, without filtering the tail region of the model's *own prediction*, the model's pretrained "knowledge" dominate the target $\pi_{tar}$ to learn, leading to a higher locality. However, the tail region of the model's *own prediction* is usually noisy, and this noise can be harmful. As shown in Tab 3, both generality and portability decreased.
> >**Typos and grammatical error.**
>
> Thank you for catching the error! We will fix this mistake in the revised draft.

---

### Decision · Program_Chairs · 2025-05-01

**Decision:**

Accept (poster)

**Comment:**

This paper presents a token-level smoothing method method called OVERTONE to address the issue of heterogeneous token overfitting (HTO) in knowledge editing tasks for large language models, aiming to improve the models' reasoning, generality, and locality with minimal computational overhead and without compromising their pre-trained capabilities.

The method has been validated through extensive experiments involving multiple editing methods, two LLMs, and diverse scenarios, demonstrating its effectiveness and versatility.

There are discussions and clarifies in the rebuttal period, with some additional experimental results.
All the reviewers give weak accept or accept recommendations.